

# Estimating Methane Emissions in the Arctic nations using surface observations from 2008 to 2019

Sophie Wittig[1,*], Antoine Berchet[1], Isabelle Pison[1], Marielle Saunois[1], Joël Thanwerdas[1], Adrien Martinez[1], Jean-Daniel Paris[1], Tochinobu Machida[2], Motoki Sasakawa[2], Douglas E. J. Worthy[3], Xin Lan[4,5], Rona L. Thompson[6], Espen Sollum[6], and Mishail Arshinov[7]

[1]Laboratoire des Sciences du Climat et de l'Environnement, CEA-CNRS-UVSQ, Gif-sur-Yvette, France
[2]Center for Global Environmental Research, National Institute for Environmental Studies, Tsukuba, Japan
[3]Environment and Climate Change Canada, Climate Research Division, Toronto, Ontario, Canada.
[4]Cooperative Institute for Research in Environmental Sciences, University of Colorado Boulder, CO, USA
[5]NOAA Global Monitoring Laboratory, Boulder, CO, USA
[6]Norsk Institutt for Luftforskning, NILU, Kjeller, Norway
[7]Independent researcher

**Correspondence:** antoine.berchet@lsce.ipsl.fr

**Abstract.**

The Arctic is a critical region in terms of global warming. Environmental changes are already progressing steadily in high northern latitudes whereby, among other effects, a high potential of enhanced methane ($CH_4$) emissions is induced. With $CH_4$ being a potent greenhouse gas, additional emissions from Arctic regions may intensify global warming in the future by positive feedback. Various natural and anthropogenic sources are currently contributing to the Arctic's $CH_4$ budget; however the quantification of those emissions remains challenging. Assessing the amount of $CH_4$ emissions in the Arctic and their contribution to the global budget still remains challenging. This is on the one hand due to the difficulties in carrying out accurate measurements in such remote areas. Besides, high variations in the spatial distribution of methane sources and a poor understanding of the effects of ongoing changes in carbon decomposition, vegetation and hydrology also complicate the assessment. Therefore, the aim of this work is to reduce uncertainties on current bottom-up estimates of $CH_4$ emissions as well as soil oxidation by implementing an inverse modeling approach in order to better quantify $CH_4$ sources and sinks for the most recent years (2008 to 2019). More precisely, the objective is to detect occurring trends in the $CH_4$ emissions and potential changes in seasonal emission patterns. The implementation of the inversion included footprint simulations obtained with the atmospheric transport model FLEXPART, various emission estimates from inventories and land surface models as well as data of atmospheric $CH_4$ concentrations from 42 surface observation sites in the Arctic nations. The results of the inversion showed that the majority of the $CH_4$ sources currently present in high northern latitudes are poorly constrained by the existing observation network. Therefore, conclusions on trends and changes in the seasonal cycle could not be obtained for the corresponding $CH_4$ sectors. Only $CH_4$ fluxes from wetlands are adequately constrained, predominantly in North America. Within the period under study, wetland emissions show a slight negative trend in North America and a slight positive trend in East Eurasia. Overall, the estimated $CH_4$ emissions are lower compared to the bottom-up estimates but higher than similar results from global inversions.



# 1 Introduction

The Arctic is an especially critical area in terms of global warming. As the near-surface air temperature has increased by approximately 3.1 °C since the 1970s, three to four times as much as the global average (AMAP, 2021; Rantanen et al., 2022), environmental changes in that region are rapidly progressing (Serreze et al., 2009; Cohen et al., 2014; Jansen et al., 2020).

Exceptional events like melting glaciers, reduction of sea ice, thawing permafrost, increasing occurrence of wildfires during summer and shortening of the snow season have already been observed increasingly frequently during the most recent years (Hassol, 2004; Stroeve et al., 2007; Walker et al., 2019). Predictions assume that, if the Arctic warming continues rising at this rate, by 2100 the temperature will have increased by 3.3 to 10.0 °C (AMAP, 2021).

Short-lived climate forcers such as methane ($CH_4$) have a significant role in this framework (AMAP, 2015). Methane is

globally the second most abundant anthropogenic greenhouse trace-gas with a radiative forcing of about 0.56 W/m$^2$ (IPCC, 2022). The rising temperatures, at the global scale and particularly in the Arctic, influence the natural $CH_4$ sources in the Arctic, which may possibly intensify local emissions in the near future (IPCC, 2022). A positive feed-back of the global - and regional - warming may therefore ensue.

Various $CH_4$ sources, both natural and anthropogenic, contribute to the Arctic methane budget. Today, the natural Arctic

methane emissions are dominated by high latitude wetlands, the extend of which is still highly uncertain however. Estimations on high latitude wetland emissions show large discrepancies. Ito (2019) concluded from a process-based modelling study that the pan-Arctic (above 60°N) wetland emissions in the 2000s to be between 10.9 and 11.4 $TgCH_4$/year. Estimates by Petrescu et al. (2010) of northern wetland emissions (defined as wetlands in regions with a yearly average temperature lower than 5°C) varied by a factor of four (between 38 and 157 Tg per year) and the corresponding regions by a factor of two (2.2 to 4.4 million

km$^2$). Uncertainties on the extend of high latitude wetland areas are, among other factors, a reason for the large variations. Other natural $CH_4$ sources occurring in this area are freshwater emissions e.g. from thermokarst lakes as well as emissions from the Arctic Ocean and biomass burning due to wildfire events in the summer months AMAP (2015). As mentioned before, natural methane emissions are anticipated to increase with rising temperatures and overall changing conditions: in the Arctic, methane net emissions could possibly be twice as high by the end of this century (Schuur et al., 2015), in part related to the

high sensitivity of $CH_4$ emissions to the state of the permafrost (Masyagina and Menyailo, 2020), and general atmospheric conditions (Chen et al., 2015). Indeed the thawing and destabilization of permafrost lead to the exposure of large carbon pools that have so far been shielded by ice and frozen soil. Permafrost thaw is expected to influence at least four ways of carbon mobilization: i) the deliberation of $CH_4$ reservoirs in the upper permafrost layers, ii) retained activity from viable methanogens as well as iii) the consumption of labile organic matter by these micro-organisms and finally iv) an increased production of $CH_4$

in the active zone (Rivkina and Kraev, 2008). Additionally, anthropogenic activities in high northern latitudes contribute to the global methane budget with an estimated amount ranging between 2 to 18 Tg $CH_4$/year (Saunois et al., 2020). These emissions are mainly caused by the exploitation and distribution of fossil fuels and are especially predominant during the winter months (Thonat et al., 2017). Currently, five Arctic nations, Russia, Canada, Norway, Greenland and the United States of America (USA), perform drilling activities in their territories and exclusive economic zones in neighbouring oceans. Decreasing the





emissions from anthropogenic sources is an effective way to limit the overall methane emissions in the Arctic region. However, with an estimated 13 % of undiscovered mineral oil and 30 % of undiscovered gas resources north of the Arctic Circle (Gautier et al., 2009), the Arctic is of significant interest for the petroleum industry regarding future drilling campaigns.

Even though the $CH_4$ observation networks in northern high latitudes have been expanded since the early 2000s, the current
stationary networks remain restricted, leaving vast areas uncovered due to the difficulties in carrying out measurements in such remote areas (Pallandt et al., 2022). Thus, obtaining accurate assessments of methane emissions in northern high latitudes remains challenging since their spatial distribution at the local scale is highly variable. Current estimations are primarily based on bottom-up studies which rely on up-scaling of local flux measurements or on process-based surface models and on emission inventories which combine emission factors with socio-economic activity data. These approaches are however subject to high
uncertainties at the regional scale since they imply statistical approximations as well as simplifications on chemical, biological and physical processes (e.g., Saunois et al., 2020).

Another approach is provided by top-down studies, in support of bottom-up products. Top-down studies optimally combine observations, provided either by ground based or satellite measurements of atmospheric $CH_4$ mixing ratios, numerical transport modelling and bottom-up emission data sets as prior emission estimates into the mathematical framework of data assimilation
to retrieve emission fluxes and their uncertainties. The so-called atmospheric inversion method is therefore useful to reduce uncertainties on bottom-up estimates (used as priors) and thus gain a better understanding on the region's methane budget. Such studies have already been implemented for high latitude regions at various scales and with regard to different sources. Inverse modelling approaches for methane emissions in the Canadian Arctic have for instance been carried out by Miller et al. (2016) (for the years 2005-2006) Ishizawa et al. (2018) (for the years 2012 to 2015), Chan et al. (2020) (for the years 2010 to
2017) and Baray et al. (2021) (for the years 2010 to 2015), for Scandinavia (Tsuruta et al., 2019, for the year 2004 to 2014), in high latitude Eurasian regions (Berchet et al., 2015, for the year 2010), for Siberian lowlands (Winderlich, 2012) and also for the whole region above 50 °North latitude (Thompson et al., 2017, for the years 2005 to 2013) and above 60 °North latitude by (Tan et al., 2016, in 2005).

In this study, we estimate methane emissions during the most recent years (2008 to 2019) through atmospheric inversion
based on available in-situ measurement data from observation sites located in the Arctic and Sub-Arctic. In order to obtain a reliable assessment, we compute a large ensemble of possible posterior emissions scenarios using different error estimations that are evaluated concerning their plausibility. The $CH_4$ emissions are subsequently analyzed with particular regards to three different questions: (i) is the available observation network sufficient to constrain all occurring $CH_4$ sources and sinks adequately? (ii) do the different $CH_4$ sources and sinks show any significant trends between the years 2008 and 2019? and (iii)
do the different $CH_4$ sources and sinks in the posterior state show any shifts in the seasonal cycle in comparison to the prior bottom-up estimates?



## 2 Methodology

To estimate the CH$_4$ fluxes in the Arctic region, a Bayesian inversion framework (2.1) based on backward simulations of the Lagrangian particle dispersion model (LPDM) FLEXPART is used (see details in Section 3.3). The inversion is based on all available observation sites in the Arctic and sub-Arctic region (see details in 2.3). Extensive sensitivity tests are carried out to evaluate the reliability of CH$_4$ estimates (see details in 2.2)

### 2.1 Inversion framework

We apply an analytical inversion which aims at explicitly and algebraically finding the optimal posterior state of a system $\mathbf{x}^a$ and the corresponding uncertainties $\mathbf{P}^a$, which are given by:

$$\begin{cases} \mathbf{x}^a &= \mathbf{x}^b + \mathbf{K}(\mathbf{y}^o - \mathbf{H}\mathbf{x}^b) \\ \mathbf{P}^a &= \mathbf{B} - \mathbf{K}\mathbf{H}\mathbf{B} \end{cases} \tag{1}$$

with $\mathbf{K}$ the Kalman gain matrix given by:

$$\mathbf{K} = \mathbf{B}\mathbf{H}^T(\mathbf{R} + \mathbf{H}\mathbf{B}\mathbf{H}^T)^{-1} \tag{2}$$

We apply the formula on a year-by-year basis in the present work. The control vector $\mathbf{x}^b$ refers to the prior knowledge on the system, in our case CH$_4$ surface fluxes from different sources (Section 3.2), but also background mixing ratios (Section 3.3.3). The observation vector $\mathbf{y}^0$ contains the available observations of atmospheric CH$_4$ mixing ratios (detailed in Section 3.1.2). The observation operator includes the transport of the emitted methane (Section 3.3.2) in the domain, the import from outside the domain (Section 3.3.3), but also, the filtering and other operations required to extract the simulated equivalents of the measurements (Section 3.3). Chemical oxidation of CH$_4$ by OH is neglected for our application (see Section 3.3). Thus, all operations in the observation operator are linear and we represent it by its Jacobian matrix $\mathbf{H}$. The linear assumption is required to write Eq. 1 and solve the Bayesian system analytically.

The error covariance matrices in the observation and control spaces, $\mathbf{R}$ and $\mathbf{B}$, define the weight of the mismatch between the modelled and the measured concentrations. $\mathbf{R}$ contains various types of errors: the error estimates of the differences between the observations and their simulated equivalents include uncertainties on the measurements, but also on the transport in the model and on the discrete representation of the continuous world by a numerical model. The dimensions of $\mathbf{R}$ are equivalent to the number of elements in the observation vector per year; it varies between 217 and 384 as observations are aggregated by station and month (see Sect. 3.1.2. The covariance matrix $\mathbf{B}$ is composed of two parts: $\mathbf{B}^S$ which accounts for the uncertainties on the prior methane fluxes and $\mathbf{B}^B$ for the uncertainties on the background mixing ratios. $\mathbf{B}^S$ has a constant size of 10164 $\times$ 10164, following the number of emission regions, emission sectors and emission periods optimized in our system (see Sect. 3.2); the dimensions of $\mathbf{B}^B$ are, again, equivalent to the number of observations per year.



Defining the error covariance matrices can be challenging since only the measurement uncertainties can be determined with certainty, using rigorous calibration procedures (e.g Sasakawa et al., 2010). On the other hand, unrealistic error estimations can drastically distort the results of the posterior state (Berchet et al., 2013). Therefore, in this study an ensemble of $(\mathbf{x}_i^a)_{i=1,500}$ and $(\mathbf{P}_i^a)_{i=1,500}$ using 500 realistic set-ups of the error matrices $(\mathbf{R}, \mathbf{B})$ is computed. The ensemble of $(\mathbf{R}, \mathbf{B})_{i=1,500}$ pairs of

matrices is described in Section 3.1.2 and in Section 3.2.2, respectively. To account for the uncertainties in the posterior state, from each vector $\mathbf{x}_i^a$, ten random variations are generated with the corresponding covariance matrix $\mathbf{P}_i^a$ following a multivariate normal distribution. Thus, we obtain a total of 5000 posterior states to assess the posterior uncertainties of the inversion.

For computational reasons, the 12-year period has been split into 12 independent 1-year inversion windows computed separately. The ensemble of 500 pairs of matrices $(\mathbf{R}, \mathbf{B})_{i=1,500}$ is generated based on a limited number of parameters independent

from the year $j$ (see Sect. 3.1.2 and 3.2.2). Therefore, for a given member $i$ of the ensemble, the yearly matrices $\{(\mathbf{R^j}, \mathbf{B^j})_i\}$ are built on the same set of underlying parameters. We then compute, for each year $j \in [2008, 2019]$, 500 independent inversions.

## 2.2 Framework evaluation

### 2.2.1 Log-likelihood of samples

Though realistically chosen (see Section 3.2.2 and Section 3.1.2), the members of the Monte-Carlo ensemble of $(\mathbf{R}, \mathbf{B})$ pairs

are not equally plausible. To further compare and aggregate statistics on the ensemble, we weight each member $i \in [1, 500]$ for each year $j \in [2008, 2019]$ by its likelihood (see, e.g., Michalak et al., 2005). It is defined by:

$$\ln p_i^j (\mathbf{R}_i^j, \mathbf{B}_i^j \mid \boldsymbol{y}_j^o, \boldsymbol{x}_j^b, \mathbf{H}_j) = -\frac{1}{2} \operatorname{tr}(\mathbf{S}_{\mathbf{R}_i^j, \mathbf{B}_i^j}^{j^{-1}} \mathbf{S}^j) - \frac{1}{2} \ln |\mathbf{S}_{\mathbf{R}_i^j, \mathbf{B}_i^j}^j| \tag{3}$$

with $\mathbf{S}^j = \boldsymbol{y}_j^o - \mathbf{H}_j \boldsymbol{x}_j^b$ and $\mathbf{S}_{\mathbf{R}_i^j, \mathbf{B}_i^j}^j = \mathbf{R}_i^j + \mathbf{H}_j \mathbf{B}_i^j \mathbf{H}_j^T$, $|\cdot|$ is the determinant operator and $\operatorname{tr}(\cdot)$ is the trace function.

The estimation of the log-likelihood provides a robust method to select the most reliable set-ups, with regards to the infor-

mation provided by the observations and ideal statistics. For a given set-up, the higher the log-likelihood, the more plausible the pair of covariance matrices. The log-likelihood estimator in a high-dimension problem like ours is extremely sensitive to any change of configuration.

The range of the log-likelihood varies between the different years, due to the variations in the number of available sites and measurements, as well as atmospheric conditions. Then, for each member of the Monte Carlo ensemble, we define the

cumulative log-likelihood as:

$$\ln p_i = \sum_{j=2008}^{2019} \ln p_i^j \tag{4}$$

We use the cumulative log-likelihood to define the most plausible posterior vector over the full period of interest from 2008 to 2019, $\mathbf{x_{max}^a}$, corresponding to the member $i_{max}$ maximizing the cumulative log-likelihood.





We also use the log-likelihood to discard the less realistic members of the Monte Carlo ensemble. To do so, the most reliable pair $i_{\mathbf{max}}^j$ of error matrices $(\mathbf{R_{max}^j}, \mathbf{B_{max}^j})$ is determined for each year $j$ separately. Then, each optimal member $i_{\mathbf{max}}^j$ for year $j$ is used on all the years of interest $j' \in [2008, 2019]$, so as to obtain corresponding cumulative log-likelihood $\ln p_{i_{\mathbf{max}}^j}$:

Since each cumulative log-likelihood $\ln p_{i_{\mathbf{max}}^j}$ includes the most reliable configuration for year $j$, the lower threshold for the

log-likelihood $\ln p_{min}$ is defined as the minimum of the 12 thus computed cumulative log-likelihood: $\min\limits_{j \in [2008, 2019]} \ln p_{i_{\mathbf{max}}^j}$. We define a sub-ensemble $\{\mathbf{x_{max}^a}\}$ whose elements have a cumulative log-likelihood greater or equal to this threshold: $\{\mathbf{x_i^a} \mid \sum_{j=2008}^{2019} \ln p_i^j > \ln p_{min}\}$. This sub-ensemble contains 274 configurations which corresponds to 2740 posterior states and is used in the following for a representative analysis of the posterior state.

### 2.2.2 Sensitivity and influence matrices

We use two other metrics to evaluate our system and the different set-ups: the influence and the sensitivity matrices. Both are calculated using the corresponding Kalman gain matrix $\mathbf{K_{max}}$ of the previously determined $\mathbf{x_{max}^a}$. The influence matrix, $\mathbf{K_{max}H}$ (defined by Cardinali et al., 2006), also called the averaging kernel (Rodgers, 2000), contains diagonal terms between 0 and 1, which represent the sensitivity of each component of $x$ to the inversion. The smaller the term $\mathbf{K_{max}H}_{r,r}$ for emissions in region $r$ is, the less constrained region $r$ is by the inversion. The sensitivity matrix $\mathbf{HK_{max}}$ (Cardinali et al., 2006) gives

the sensitivity of the inversion to a change in one component of the observation vector. An observation with a high sensitivity brings strong constraints on the inversion. The weight of each station in the inversion can be computed by summing up the corresponding diagonal elements of $\mathbf{HK_{max}}$. The trace of these two matrices also gives the "degrees of freedom for signal" (Wahba et al., 1994; Cardinali et al., 2006), while the number of observations minus this number gives the "degree of freedom for noise". This extra criterion informs on how much observations are used to constrain fluxes (and background mixing ratios).

### 2.3 Area and period of interest

The area of interest, shown in Figure 1a, for this study regarding the quantification of the methane fluxes includes the Arctic and Sub-Arctic, with the southern boundary being roughly the southernmost border of the taiga. For the implementation of the inversion, only observation sites within the area of interest have been included in this study. To represent concentrations at these sites as properly as possible, we simulate the influence of fluxes from the area of interest, but also from a buffer region

from above 30 °N (see Sect. 3.3.1). Even though Arctic fluxes may influence observation sites in the buffer region, we do not include them in this study due to the increased computational costs this would induce; a future work may inquire into the impact of using as many stations as possible.

The region above 30 °N is subsequently divided into sub-regions in order to better detect local differences. However, the sub-regions should not be too small and numerous, due to the limitation of available observations for constraining those

areas. A more detailed description of the selected observation sites (indicated with white stars in Figure 1b) can be found in Section 3.1.1. The sub-regions of this study are therefore selected following the proposition of the Regional Carbon Cycle Assessment and Processes (RECCAP; Ciais et al., 2022) which results in 121 regions within the area of interest (Figure 1b).





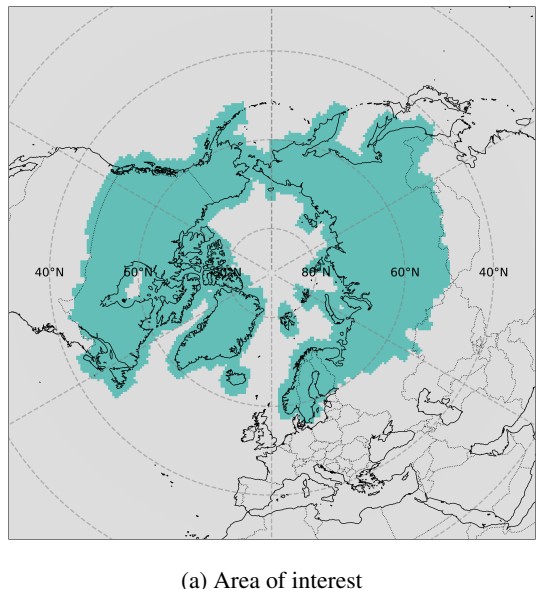

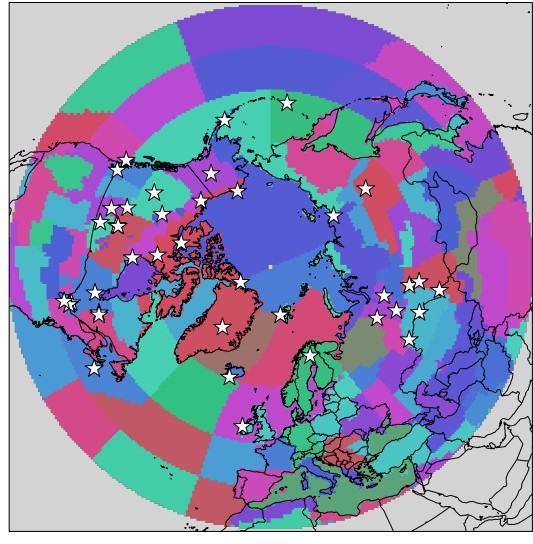

(a) Area of interest
(b) RECCAP regions

**Figure 1.** Area of interest (left) and RECCAP regions above 30 °N (right). Measurement sites, listed in Table S1, are indicated with white stars.

The time period of interest is from 2008 to 2019. For the following years, no measurements were available for the majority of the measurement sites by the time this study was implemented.

The atmospheric sites in the area and time period of interest and the available observations are described in Section 3.1. $CH_4$ emissions in this area are described in Section 3.2.

## 3 Material

### 3.1 Atmospheric observations

#### 3.1.1 Site description

For this study, both quasi-continuous measurements (35 observation sites providing hourly measurements) and discrete measurements (6 observation sites providing task samples two to four times a month) are used. The stations are exclusively located in 7 Arctic nations (Canada, Russia, Finland, Norway, Iceland, Greenland and the USA), except for one site in Ireland, used to constrain air masses from the Atlantic Ocean. The operators of these stations are Environment and Climate Change Canada, the Japan–Russia Siberian Tall Tower Inland Observation Network (JR-STATIONS from NIES; Sasakawa et al., 2010), the U.S National Oceanic and Atmospheric Administration Global Monitory laboratory (NOAA-GML; Dlugokencky et al., 2020) and the Finnish Meteorological Institute (FMI; Hatakka et al., 2003; Aalto et al., 2007). The stations with their trigram identifications are shown in Figure 2.





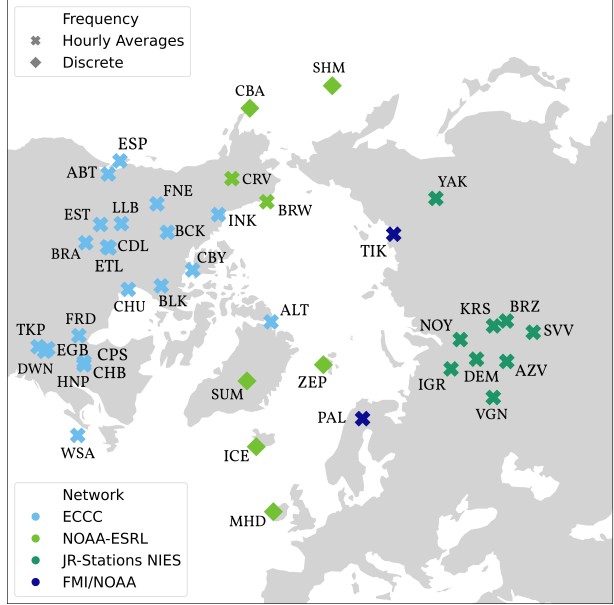

**Figure 2.** Map of the selected observation sites. Crosses indicate quasi-continuous, diamonds discrete measurements. Different network operators are marked with different colours.

All the measurement sites are subsequently described briefly sorted by their network operators. A summary of each station's characteristics is furthermore provided in the supplements (Table S1).

**ECCC**

The ECCC established its first two $CH_4$ measurement stations (ALT and FRD) in the end of the 1980s and has expanded

its network to 22 sites to this date, 12 of them being located in the Arctic or Sub-Arctic. Alert (ALT) is often referred to as an Arctic background site since it is located remotely from any major methane emission sources on the northeastern tip of Ellesmere Island in Nunavut where the land is covered with snow for approximately ten month a year. Two additional sites are installed in Nunavut in slightly more southern latitudes: Cambridge Bay (CBY) and Baker Lake (BLK). The latter is located in the Arctic Tundra around 320 km from Hudson Bay surrounded by small lakes whereas CBY lies on the south-east coast of

Viktoria Island close to the largest port of the Northwest Passage of the Arctic Ocean.

The measurement site Inuvik (INK) was established in the Arctic Tundra of the Northwest Territories in the east channel of the Mackenzie Delta. Further inland in the same Canadian province lies the station BCK, 10 km from the town Behchoko and surrounded by mixed forests, lakes and ponds.

Three of the ECCC sites are located in British Columbia. FNE, which is located close to small town Ford Nelson in the

15 Taiga, lies at the southern fringe of the Canadian permafrost region. Estevan Point (ESP) is located on the coast of the Pacific





Ocean and surrounded by woodlands. The measurement station Abbotsford (ABT) lies close to the US border 80 km from Vancouver, the largest city and main economic area in British Columbia.

The two sites in the province Alberta are LLB at the lake Lac La Biche in a region of peatlands and forest and Esther (EST) which lies in the open prairie with plenty of cattle ranches close by.

Two measurement stations are established in Saskatchewan. East Trout Lake (ETL) in the center of the province lies at the southern edge of a boreal forest region and Bratt's Lake (BRA) in the Canadian prairie.

Churchill (CHU) is located Manitoba, north of the largest continuous boreal wetland region in North America on the west coast of Hudson Bay.

Four of the sites in the province Ontario (EGB, DWN, HNP, TKP) are located relatively close to each other in the Mixedwood
Plains Ecozone. Downsview (DWN) and Hanlan's Point (HNP) are urban stations in in the north of Toronto and on the Toronto Islands in Lake Ontario, respectively. Egbert (EGB) lies around 80 km from Toronto close to a rural village. The south most site Turkey Point (TKP) is located at Lake Erie in a woodland area. Further north in Ontario lies the station Fraserdale (FRD) in the boreal forest with extensive wetland coverage in the surrounding.

The two sites located in Quebec, Chapais (CPS) and Chibougamau (CHB), are likewise established close to each other in an
area dominated by boreal forest with many lakes.

Finally, the observation site Sable Island (WSA) is on a remote island in the North Atlantic Ocean, 175 km from the mainland. The island is uninhabited by people and covered with grass and low-growing vegetation.

**NOAA-GML**

The two continuous measurement stations operated by NOAA-GML are Barrow (BRW) and CARVE (CRV) in the USA
(Dlugokencky et al., 2020). Methane measurements in BRW started in the late 1980s. The site is located in northern Alaska on the junction of the Chukchi and Beaufort Seas and the surrounding landscape is characterized by thermokarst lakes. The CRV tower is located in boreal Alaska with a surrounding landscape defined by evergreen forest, shrubland and some areas of woody wetlands (Karion et al., 2016).

The six discrete measurement sites operated by NOAA-GML are ZEP, SUM, ICE, MHD, CBA and SHM. The Zeppelin
Observatory (ZEP) is located near the village Ny-Aalesund, which is surrounded by mountains and glaciers, on the island Spitsbergen. From 2017, ZEP observations are available as continuous data via the ICOS Carbon portal (Lund Myhre et al., 2022), but we did not include them as such to avoid perturbing the interpretation of the results for the last years. The sampling site Summit (SUM) was established on the Greenland Ice Sheet and is the highest measurement site of the Arctic Circle. Storhofdi (ICE) lies in the South of Iceland at the top of a small cape with grassy slopes and cliffs to the sea close by. The
sample site Mace Head (MHD) is located at the west coast of Ireland in a wet and boggy area. The surrounding landscape is and characterized by small hills covered with grasses and sedges with many exposed rocks. At the southern tip of the Alaska Peninsula nearby the coast lies the measurement site Cold Bay (CBA) within a wet tundra ecosystem consisting of a variety of sedges and grasses. Finally, the station SHM is located on the island Shemya, which belongs to a cluster of small islands southwest of Alaska.



**JR-STATIONS**

The four JR-STATIONS have been installed by NIES in 2004. Three are located in the Russian taiga forest surrounded by wetlands: Demyanskoe (DEM), Karasevoe (KRS), Noyabrsk (NOY). Additionally, one station was installed in a small town close to the Ob river with around 10.000 inhabitants, likewise surrounded by wetlands.

The network has been extended by five stations in the upcoming years incorporating different biomes. Three towers have been placed in steppe regions. Azovo (AZV) and Vaganovo (VGN) are located in the immediate vicinity of highly populated cities whereas the SVV-tower (Savvushka) is installed near a small village. Additionally, one tower is located in the middle of the taiga surrounded by boreal forest (Berezorechka, BRZ) and lastly, the YAK-tower was placed close to Yakutsk in the East Siberian Taiga (Sasakawa et al., 2010; Belikov et al., 2019). However, not all of the JR-stations are currently still in operation:

the dates of beginning and end of operation are indicated in Table S1. Since the towers are provided with two to four different sampling heights up to 85 magl, only the measurements from the highest inlet are used in this study. The $CH_4$ measurements are reported on the NIES-94 scale and have been converted to the NOAA 2004 scale following Zhou et al. (2009).

**FMI/NOAA**

The Finnish station Pallas (PAL) is located close to the northern edge of the Scandinavian boreal zone with a surrounding terrain

of wetlands, lakes and patches of forest (Hatakka et al., 2003; Aalto et al., 2007). PAL data are available as FMI GAW CH4 data from 2004 onwards at the World Data Center for Greenhouse Gases (WDCGG). PAL data from 2017 are also available from the ICOS carbon portal Hatakka and RI (2022). Like PAL, the site Tiksi (TIK) is operated by the FMI in cooperation with NOAA-GML and is installed on the shore of the Laptev Sea on the Lena river delta (Uttal et al., 2013, 2016).

### 3.1.2 Data selection and observation uncertainties

In regional inversions, concentration peaks carry a large part of the information content on local to regional fluxes. However, transport can be erroneous and simulated peaks can be shifted in time compared to observed ones, although the magnitude can be well represented. Such errors heavily penalize Bayesian inversions, so we decided to aggregate observations at the monthly scale. This focuses the inversion on emission trends and seasonal cycles.

In the observation vector $\mathbf{y}^0$ (Section 2.1), we use the monthly averages of the available $CH_4$ atmospheric measurements

at each site. When hourly quasi-continuous data was available, only measurements between 12:00 and 16:00 local time are selected, assuming a well-mixed boundary layer, which is better simulated by the model (Section 3.3). The discrete observations are not filtered by the time of day the measurement was taken. However, the data sets contain several measurement outliers, mostly strong concentration peaks related to local emissions, difficult to simulate with our transport model. We excluded such peaks from the observations used for the inversion if they differed more that 5 % (or 100 ppb) from the monthly average.

Depending on the measurement site, between 8 and 20 % of the observations are discarded this way.





Due to the discontinuity of measurement availability, the size of $\mathbf{y}^0$ for one year varies between 217 (2008) and 384 (2018). The number of observations per year used for the inversion (and thus the size of $\mathbf{y}^0$) can be found in Table S2. All the selected observations with the corresponding daily $CH_4$ concentrations are shown in Fig. 3.

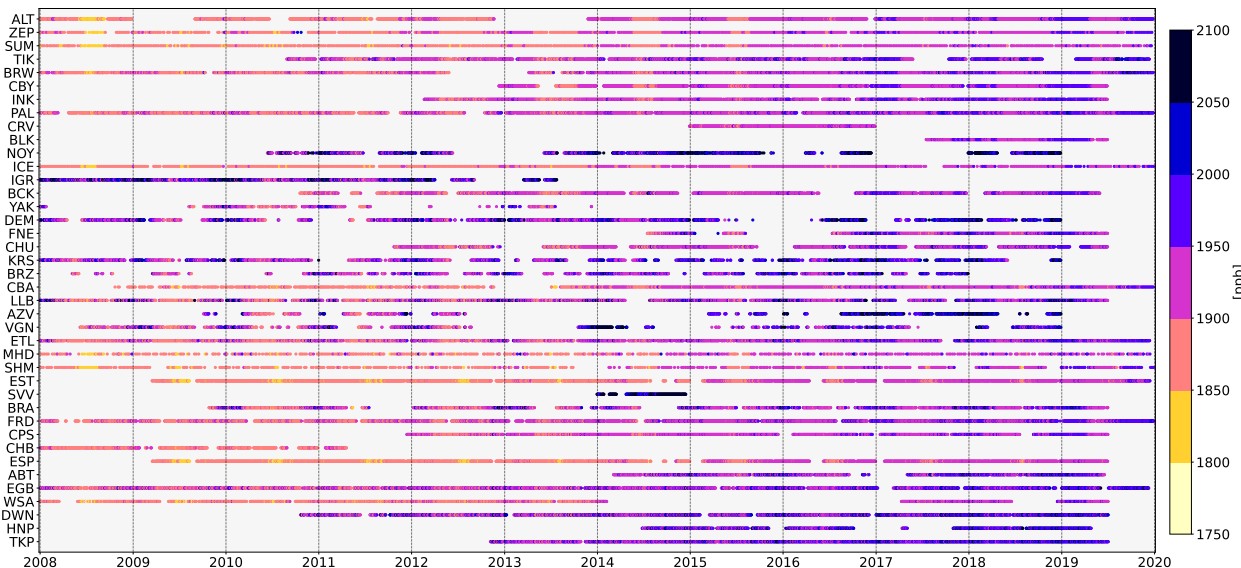

**Figure 3.** Average daily methane concentration at each station. The observation sites are sorted by latitude.

The corresponding uncertainties on the observations are specified in the diagonal error covariance matrix $\mathbf{R}$, of which an
ensemble of 500 set-ups is generated (Section 2.1).

To generate a large number of different error set-ups, the first step consists in obtaining an estimate of the uncertainty for each station $s \in [1, 41]$ and each year $j \in [2008, 2019]$ which serves as a reference point. This is done by computing the differences between the monthly mean of the measured and corresponding modelled mixing ratios (see Section 3.3) in absolute values:

$$\mathbf{\Delta}_{m,j}^s = |\mathbf{y}_{m,j}^{model} - \mathbf{y}_{m,j}^{obs}| \tag{5}$$

with $\{m | 0 \leq m \leq 12]\}$ one month of a given year $j$.

Then, the standard deviation of the ensemble of 12 monthly differences is computed for each year:

$$\sigma_{s,j}^R = \sqrt{\frac{1}{12} \sum_{m=1}^{12} \left( \Delta_{m,j}^s - \overline{\Delta_{m,j}^s} \right)^2} \tag{6}$$

with $\overline{\Delta_{m,j}^s} = \frac{1}{12} \sum_{m=1}^{12} \Delta_{m,j}^s$. In the few cases when only one observation is available for a given station and a given year, no standard deviation can be computed so that the single difference between the modelled mixing ratio and measurement is used
directly.





The obtained errors per station and year are subsequently varied following a log-normal distribution with $\sigma_{s,j}^R$ as its mode. This error distribution is chosen to include only a few very high outliers in the ensemble. To implement a log-normal distribution, a standard deviation $\sigma_{i,random}^R$ must be provided which is constant for each element $i$ of the ensemble $i \in [1,500]$. Thus, the random observation error for each station $s$ is equal for all months within one year, however varies between the different years of one element $i$ of the ensemble. To ensure that the values of the observation errors do not vary to an unrealistic extent, a minimum of 0.5 ppb and a maximum of 150 ppb are set.

Finally, the elements of the diagonal of one error covariance matrix $\mathbf{R}_i^j(k,k)$ for $k \in \{s \in [1,41]\} \times \{m | 0 \le m \le 12\}$ and $i \in [1,500]$ are defined as the variances $\left(\sigma_{s,j}^{R,i}\right)^2$ and the non-diagonal elements are zero.

Figure 4 shows an example of the frequency distribution of the observation errors at one of the selected sites, $s =$INK, for the year $j =$2012. The mode and therefore the reference point of the observation error for this year and station is around 8 ppb. To give an idea about the general magnitude of the computed uncertainties, the average of $\sigma_{s,j}^R$ over all the stations $s$ including all years $j$ in the period of interest is around 18 ppb.

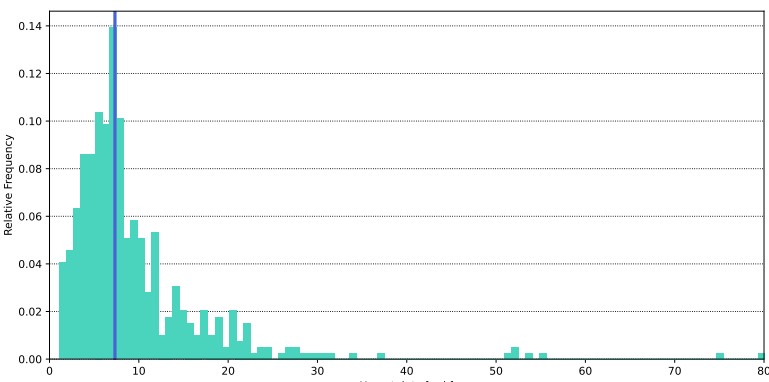

**Figure 4.** Frequency distribution of the 500 random observation errors $\left\{\sigma_{s,j}^{R,i}\right\}$ for $i \in [1,500]$ at $s =$INK for the year $j$=2012. The blue line marks the mode $\sigma_{INK,2012}^R$.

## 3.2 Prior Emissions

### 3.2.1 Emission scenarios

The emissions used as prior information are based on a set of various inventories and models. The different methane sources and sinks are described in Table 1 with there respective temporal resolution in the prior. The natural CH$_4$ sources include emissions from wetlands, the Arctic Ocean and geological sources. Natural methane emissions caused by biomass burning due to wildfire events are combined with anthropogenic biofuel activities for simplification. Since emissions caused by termites are negligible in the Arctic, they are not taken into account in this study. For the CH$_4$ sink, soil oxidation is included as negative emissions. To reduce the number of sectors to optimize, emissions related to the exploitation and distribution of mineral oil





**Table 1.** *Methane sources and sink taken into account in the prior emissions. The share of the global emissions of each source is based on the average fluxes between 2008 and 2019. For data sets with inter-annual differences, the range between the lowest and highest emissions is given. EDGARv6 is described in Crippa et al. (2021) and GFED4.1 in RANDERSON et al. (2017).*

| Type | Source | Reference | Emissions globally (Tg CH$_4$.yr$^{-1}$) | Emissions area of interest (Tg CH$_4$.yr$^{-1}$ / % of global emissions ) | Temporal resolution |
|---|---|---|---|---|---|
| Natural | Wetlands | Poulter et al., 2017 | 179.95 | 44.80 / 24.9 | monthly climatology |
| | Ocean | Weber et al., 2019 | 11.48 | 3.02 / 26.3 | constant |
| | Geological | Etiope et al., 2019 | 36.67 | 7.66 / 20.9 | constant |
| | Soil Oxidation | Ridgewell et al., 1999 | -37.88 | -4.74 / 12.5 | monthly climatology |
| Combined | Biomass and biofuel burning | GFED4.1 | 24.28 - 34.69 | 1.87 - 4.00 / 10.1 | monthly with interannual variability |
| | | EDGARv6 | | | |
| Anthropogenic | Mineral oil & gas | EDGARv6 | 102.26 - 126.90 | 14.70 - 17.83 / 14.6 | interannual variability |
| | Waste & Agriculture | EDGARv6 | 216.38 - 236.49 | 8.58 - 8.77 / 3.8 | interannual variability |
| Total | | | 542.80 – 587.74 | 75.89 – 81.28 / 17.3 | |

and gas have been combined to a single data set. The same applies to the emissions from agricultural activities and waste management.

For the natural sources as well as the soil sink, monthly climatological data sets are used for the whole period so that their total fluxes do not differ between the years 2008 to 2019. The emissions from anthropogenic sources vary between the different years covered in this study, following the EDGARv6 emission trends (Crippa et al., 2021). Emissions caused by fossil fuel activities generally increase between 2008 (15.96 Tg/year) and 2019 (17.31 Tg/year) though the highest annual emissions occur in the years 2014 and 2015. Methane emissions from agricultural activities and waste management also increase slightly throughout the period of interest however just by less then 0.18 Tg/year. The combined biomass burning scenario also shows some inter-annual variability though without any apparent tendencies. The lowest annual emissions occur in 2009 (1.87 Tg CH$_4$) and the highest in 2012 (3.99 Tg CH$_4$).



At the intra-annual scale, in contrast to the other natural $CH_4$ sources, the wetland scenario has a clear seasonality in the Arctic with higher emissions during the summer months. According to the data set used for this study, the highest wetland emissions occur in August (10.72 Tg $CH_4$/month) and the lowest in January (0.04 Tg $CH_4$/month). The soil methane oxidation has a seasonal pattern symmetric to the wetland emissions with the maximum uptake taking place in August (-1.02 Tg $CH_4$/month)

and a minimum in January (-0.01 Tg $CH_4$/month). The combined biomass burning scenario shows a small seasonal variability with predominantly higher emissions during the summer. Between 2010 and 2016, the highest monthly $CH_4$ emissions occur in July and from 2017 to 2019 the peak emissions take place in August. Hereby, the maximum of the methane emissions ranges between 0.49 Tg $CH_4$/month (2009) and 1.91 Tg $CH_4$/month (2017). The first two years within the period of interest do not fall into this seasonal pattern with increased $CH_4$ fluxes during the summer months. Regarding the anthropogenic methane

emissions, the agricultural and waste management fluxes also show a seasonal pattern with increased emissions during the summer. According to the inventory, the emissions are highest in June (around 0.80 Tg $CH_4$/month) and lowest in January and December (around 0.67 Tg $CH_4$/month). The methane emissions from oil and gas exploitation and distribution are nearly constant over the course of each year with a maximum variation of 0.1 Tg $CH_4$/month.

### 3.2.2 Prior uncertainties

As for the observation error, the elements of the prior error matrix $\mathbf{B}$ are obtained from a random sampling. The covariance matrix thereby contains both the uncertainties on the prior fluxes $\mathbf{B}^S$ and the uncertainties on the background mixing ratios $\mathbf{B}^B$. In the following, only the methodology of the random sampling of the prior errors is explained, the details on $\mathbf{B}^B$ are described in Section 3.3.3.

For each $CH_4$ source or sink $S$, the mode $\sigma^S$ is set following Baray et al. (2021):

– 50 % for $S$ =anthropogenic emissions

  – 60 % for $S$ =wetland emissions

  – 100% for $S$ =other natural sources and soil oxidation

A random sampling following a log-normal distribution with $\sigma^S$ as its mode results in an ensemble of 500 prior errors per source or sink $\left\{ \sigma_i^{B,S} \right\}_{i \in [1,500]}$. These random errors remain identical for each region $r$ and month of year $m$ per element $i$ of

the ensemble. Exemplarily, figure 5 shows the frequency distribution of the random prior errors for $S$ =wetlands emissions of all the set-ups.

Finally, the elements of the diagonal of one error covariance matrix $\mathbf{B}_i^S(k,k)$ for $k \in \{S \in [1,7]\} \times \{r \in [1,121]\} \times \{m \in [1,12]\}$ are defined as the variances $\left( \sigma_i^{B,S} \right)^2$. Hereby, $\mathbf{B}_i^S$ is identical for each year.





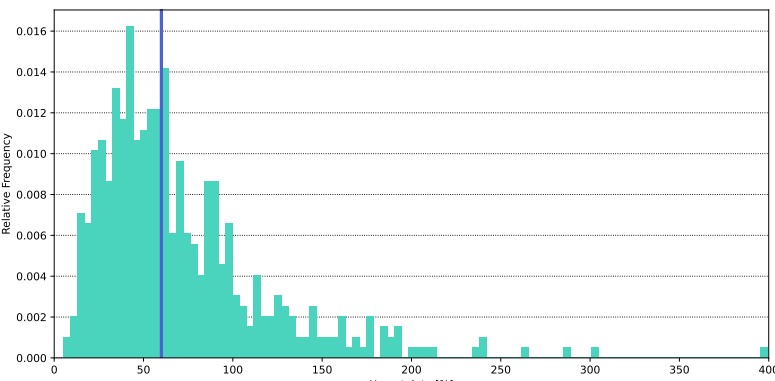

**Figure 5.** Frequency distribution of the 500 random prior errors $\left\{\sigma_i^{B,S}\right\}_{i\in[1,500]}$ for $S =$wetland emissions. The blue line marks the mode $\sigma^{wetlands}$

The off-diagonal elements $(\sigma_{m,n})^2$ with $m$ the row and $n$ the column of the corresponding matrix $\mathbf{B}_i^S$ are determined by applying spatial and temporal correlations. $\mathbf{B}_i^S$ is hereby a symmetrical matrix so that $(\sigma_{m,n})^2$ is identical to $(\sigma_{n,m})^2$.

The off-diagonal errors are computed as follows:

$$(\sigma_{m,n})^2 = (\sigma_{n,m})^2 = \left(\frac{(\sigma_{m,n=m})^2 + (\sigma_{m=n,n})^2}{2}\right) \times \exp(-\frac{\Delta t}{t_{corr}}) \times \exp(-\frac{\Delta d}{d_{corr}}) \tag{7}$$

with $\Delta t$ the temporal difference between the rows/columns $m$ and $n$ and $\Delta d$ the spatial difference referring to the centres of the corresponding regions. For the spatial correlation $d_{corr}$ a distance of 500 km is used and the temporal correlation $t_{corr}$ has a fixed value of one week.

### 3.3 Modelled CH$_4$ mixing ratios

As mentioned in section 2.1, the simulated equivalents to the observations are included in the observation operator $\mathbf{H}$. In this case, $\mathbf{H}$ consists as well of the monthly CH$_4$ mixing ratios sectioned into sub-regions and sectors as of the monthly averages of the background mixing ratios by station. $\mathbf{H}$ is hereby linear since only emissions and transport of CH$_4$ are taken into account. The oxidation of methane by hydroxyl radicals (OH) is neglected since the life time of CH$_4$ is ≈9 years (Prather et al., 2012) and the air masses remain in the domain up to 2 months (Berchet et al., 2020).

#### 3.3.1 Transport model set-up

The modelled CH$_4$ mixing ratios were obtained by using the Lagrangian atmospheric transport model FLEXPART (FLEXible PARTicle) version 10.3 (Stohl et al., 2005; Pisso et al., 2019). This model simulates numerous trajectories of infinitesimally small air parcels, called particles, and can be used either forward or backward in time. FLEXPART is an offline model that is driven by meteorological data from the European Centre for Medium-range Weather Forecast (ECMWF) ERA5 (Hittmeir et al., 2018) with 3-hourly intervals and 60 vertical layers. ECMWF data are retrieved and formatted using the FLEX-extract



toolbox (Tipka et al., 2020). In this study, 2000 particles are released at each observation site and time stamp (receptor) and followed 10 days backwards in time. The horizontal resolution is $1\,^\circ \times 1\,^\circ$, which is quite commonly used for inverse modelling set-ups using Lagrangian particle dispersion models in high northern latitudes (e.g. Thompson et al., 2017; Ishizawa et al., 2018).

### 3.3.2 Source contribution

By sampling the near-surface residence time of the various backward trajectories of the particles the source-receptor sensitivity matrices, also called footprints, of each observation site can subsequently be determined. These footprints define the connection between the fluxes discretised in space and time and the change in concentrations at the receptor (Seibert and Frank, 2004). To finally obtain a time series of modelled $CH_4$ mixing ratios, a time series of footprints is integrated with discretised methane emission estimates. Here, monthly averages of the footprints of each receptor are used to determine the mixing ratios for each sector (see Table 1 in Section 3.2.1) and sub-region (see Figure 1b in Section 2.3).

The magnitude of the thus obtained total $CH_4$ mixing ratios, including all methane sources and the soil sink, ranges roughly between 3 ppb and 90 ppb depending on the month of the year and location of the observation site and the average standard deviation is around 14 ppb.

### 3.3.3 Background mixing ratios and uncertainties

Since $CH_4$ has a much longer lifetime than the released virtual particles, the previously obtained concentrations only display short-term fluctuations at the receptors. Therefore, in order to obtain a direct comparison to the measurements, the background mixing ratio needs to be taken into account.

The background mixing ratios are calculated by combining a $CH_4$ concentration field as initial condition with the FLEX-PART backward simulations nudged to the observations of the corresponding site (e.g. Thompson and Stohl, 2014; Pisso et al., 2019). The background thus obtained represents the average of the mixing ratios in the grid cells where each particle trajectory terminated 10 days before the observation. The initial concentration field is provided by the Copernicus Atmospheric Monitoring Service (CAMS): a $CH_4$ mixing ratio field from CAMS global reanalysis EAC4 (ECMWF Atmospheric Composition Reanalysis 4) with 60 vertical layers, a 3-hourly temporal and a $0.75\,^\circ \times 0.75\,^\circ$ spatial resolution has been used (Inness et al., 2019). The implementation used for the obtaining the background mixing ratios is provided by the Community Inversion Framework (CIF) (CIF; Berchet et al., 2021b).

The thus computed background mixing ratios show a gradual increase over the period of interest with mean annual concentrations over all sites ranging between 1842 ppb (2008) and 1974 ppb (2019). At intra-annual scale, the monthly background mixing ratios vary from the corresponding annual average by around 8 %. Figure S5 (supplements) shows the average background mixing ratios at each station as well as their average standard deviation.

As stated previously, the background is the major share of the total modelled mixing ratios and, in this study, makes up approximately 97.6 % at continental observations sites and 99.5 % at stations located remotely. A summary of the proportion of source contribution and background mixing ratios for each station can be found in Table S4 in the supplements.



As mentioned before (e.g. Section 2.1), the uncertainties on the background mixing ratios $\mathbf{B}^B$ are included in the error covariance matrix $\mathbf{B}$. In contrast to the uncertainties on the prior emissions $\mathbf{B}^S$, which are given by region, month and $CH_4$ source/sink, the uncertainties on the background mixing ratios are given by observations site and month. Therefore, the size of $\mathbf{B}^B$ is equivalent to the number of available observations per year.

The elements of $\mathbf{B}^B$ are composed in a similar manner as the elements of $\mathbf{R}$ (Section 3.1.2), by first computing a reference error for each station and year and varying these values randomly to obtain and ensemble of 500 set-ups.

In this case, the standard deviations of the monthly background mixing ratios $\mathbf{y^{back}_{s,m,j}}$ per station $s \in [1, 41]$ and year $j \in [2008, 2019]$ serve as reference errors:

$$\sigma^B_{s,j} = \sqrt{\frac{1}{12} \sum_{m=1}^{12} \left( y^{back}_{s,m,j} - \overline{y^{back}_{s,m,j}} \right)^2} \tag{8}$$

with $\overline{y^{back}_{s,m,j}} = \frac{1}{12} \sum_{m=1}^{12} y^{back}_{s,m,j}$ and $m \in [1, 12]$.

Subsequently, the computed errors per station are varied following a log-normal distribution with a mode of $\sigma^B_{s,j}$. Again, in order to achieve a log-normal distribution, a random standard deviation $\sigma^B_{i,random}$ must be set which is consistent per element $i \in [1, 500]$ of the ensemble. Similar to the observation errors, this means that each observation site $s$ has identical values of background errors for every month $m$ within one year but each station may have unequal errors for the different years $j$ of one

element $i$ of the ensemble. The lower and upper limits of the background mixing ratio uncertainties are hereby 0.5 ppb and 150 ppb.

The diagonal elements of one error covariance matrix $\mathbf{B}^{B,j}_i(k,k)$ for $k \in \{s \in [1, 41]\} \times \{m|0 \le m \le 12\}$ and $i \in [1, 500]$ are finally defined as the variances $\left( \sigma^{B,i}_{s,j} \right)^2$.

Other than the observation error covariance matrix $\mathbf{R}$, $\mathbf{B}^B$ is not a diagonal matrix and the non-diagonal elements are defined

by applying correlations in space and time. The computation of the non-diagonal errors $(\sigma_{m,n})^2$ with $m$ the corresponding row and $n$ the corresponding column of the symmetrical matrix $\mathbf{B}^B_i$ is similar to the implementation of correlations for the prior error covariance matrices $\mathbf{B}^S_i$ Section 3.2.2:

$$(\sigma_{m,n})^2 = (\sigma_{n,m})^2 = \left( \frac{(\sigma_{m,n=m})^2 + (\sigma_{m=n,n})^2}{2} \right) \times \exp(-\frac{\Delta t}{t_{corr}}) \times \exp(-\frac{\Delta d}{d_{corr}}) \tag{9}$$

with $\Delta t$ the temporal difference between the rows/columns $m$ and $n$ and $\Delta d$ the spatial difference referring distance between

the two corresponding measurement sites. The correlation lengths are $d_{corr} = 500$ km for spatial correlations and $t_{corr} =$ one week for temporal correlations.





## 4    Results

### 4.1    Performance of the inversions in the observation space

To evaluate the performance of the inversion, the prior and posterior CH$_4$ mixing ratios are compared to the observations. Figure 6 shows the Taylor diagrams indicating the Pearson correlation coefficient to determine similarities between the observations and simulations as well as the normalized standard deviation (SD) displaying how well the variability of the modelled mixing ratios is captured. Thus, a shorter distance to the reference point indicates a closer fit to the measured mixing ratios. In Figure 6, we split results for the full data set and de-trended data. The performance of the simulations for the full data set is mostly driven by the long-term trend. The de-trended data exhibits the performance in terms of seasonal cycle.

In general, and as expected, the posterior results show a better agreement with the observations compared to the prior mixing ratios of the corresponding observation site. This is more distinctive for the trended (Figure 6a to Figure 6c) than for the de-trended time series (Figure 6d to Figure 6f), although in both cases the majority of the posterior mixing ratios is closer to the measurements than the prior ones. This confirms that the climatological priors are not realistic and the inversion can realistically improve the flux trends. Both the normalized standard deviation and the correlation coefficient should ideally be close to 1. The prior trended SD range between 0.19 and 1.62 and the correlation coefficients between 0.20 and 1.09. For the posterior results the values lie between 0.19 and 1.00 (standard deviation) and 0.29 and 1.0 (correlation coefficient). Regarding the de-trended time series the normalized SD lies between 0.19 and 2.61 (prior) and 0.02 and 0.99 (posterior) and the correlation coefficient ranges between 0.20 and 1.41 (prior) and 0.10 and 1.00 (posterior).

The improvement in the posterior results is quite evident for observation sites which are remote from methane emission sources, such as ALT or ZEP (Figure 6a), where the posterior results are nearly equal to the observations. H ere, the standard deviations have a maximum deviation of 0.10 from the observations point whereas the difference between the correlations is ≤ 0.02. It is however noteworthy that the prior CH$_4$ concentrations show already a good agreement with the observations at those remote stations which are often referred to as background observation sites. This good fit can be explained by the fact that background mixing rations are computed using global mixing ratio fields generated by systems optimized using these same remote sites.

A much larger improvement can be observed at sites close to the North American coast such as INK, BLK, and CHU (Figure 6c). In general, the majority (8 out of 11) of the measurements of the coastal stations has a lower standard deviation than their modelled equivalents which implies that the variability of the modelled mixing ratios is overestimated. The magnitude of the prior modelled CH$_4$ mixing ratios is overall higher than the measurements at North American observation sites in high northern latitudes (up to approximately 80 ppb; and on average 50 ppb).

The simulations of continental observation sites further South in North America (e.g. BRA) as well as continental sites in Russia such as NOY and IGR (Figure 6b) show, in general (8 out of 12 sites), a normalized standard deviation which is lower than the observations. This indicates an underestimation of the variability in the simulated CH$_4$ mixing ratios, both in the prior as in the posterior results. However, the correlation with the observations could still be improved by the inversion.







(a) Remote sites, trended      (b) Coastal sites, trended      (c) Continental sites, trended

(d) Remote sites, de-trended      (e) Coastal sites, de-trended      (f) Continental sites, de-trended

**Figure 6.** Examples of Taylor diagrams for various site categories. Raw (upper row) and de-trended (lower row) mixing ratios over the whole period of interest. Prior simulated mixing ratios are indicated with circles, posterior with diamonds.

The only observation site where the posterior results show less agreement with the corresponding measurements is ABT (Figure 6b) where, in contrast to most other sites in North America, the observations are significantly higher than the simulated CH$_4$ mixing ratios by up to approximately 100 ppb. Local fluxes (mostly from urban environments) and complex topography (mountain range surrounding the flat area around Vancouver, Canada) are likely to influence the observations at this site and are ill represented by the model at a coarser resolution. A higher transport resolution and finer scale inversion regions could solve this issue in a future study; stations too close to urban centers and in too complex topographical configurations could also be discarded altogether to pan-Arctic studies focusing on large scale patterns.





## 4.2 Distribution of information in the inversion system

### 4.2.1 Impact of observations on the inversion system

To further analyze the network efficiency, the sensitivity matrices **HK** (see Section 2.2.2) are calculated for each year and averaged over the whole period of interest (Figure 7). The percentages indicate how much of the theoretically available obser-
vations at each site are actually used by the inversion. The observation sites which are located remotely from any other stations, mostly along the Arctic, Atlantic and Pacific oceans shores, show values of almost 100 % which means that the information provided by the measurements are almost entirely used, mostly to constrain background concentrations. This is confirmed by the amplitude of the background at these sites, as shown in Figure S5, where the ratio between the standard deviation of the simulated signal from the background and from emissions has been computed and show similar patterns than the sensitivity to
observations. In areas where the observation network is much denser (e.g. in the Southeast of Canada, and in a lesser extent in Siberian lowlands), most observation contribute for less than 50 % to the inversion. Lower constraints in dense continental areas are caused either by redundant constraints by neighbouring sites in the same emission areas and/or higher noise due to transport errors from nearby emissions. The latter has the largest impact if the site is located close to $CH_4$ emission sources.

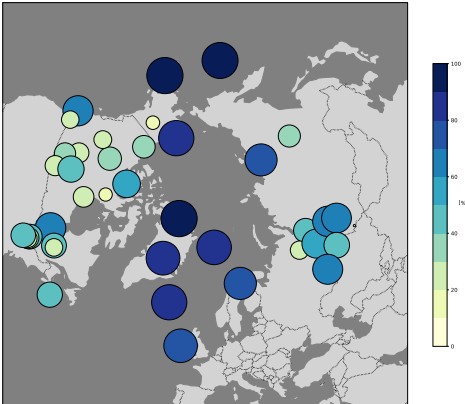

**Figure 7.** Sensitivity of the inversion to observation sites. Larger and darker circles indicate a higher usage of available observations by the inversion.

### 4.2.2 Noise and information content in the inversions

The trace of the influence matrix $tr(\mathbf{KH})$ (equal to the trace of the sensitivity matrix, $tr(\mathbf{HK})$) indicates how much noise is contained in the provided observations, and how the information content is used by the inversion (Section 2.2.2). The closer the value of $tr(\mathbf{KH})$ is to the number of available observations, the more useful is each given observation for the inversion. Furthermore, the ratio between the number of observations used to constrain the emissions and used to constrain the background mixing ratios can be determined by separately calculating $tr(\mathbf{KH_{emis}})$ and $tr(\mathbf{KH_{back}})$, using only the



corresponding elements of **KH**. The obtained traces for each year are given in Table S2 (see supplements) and Figure S3 shows the ratio between $tr(\mathbf{KH_{emis}})$ and $tr(\mathbf{KH_{back}})$.

In total, $tr(\mathbf{KH})$ ranges between approximately 60 and 75 % of the number of available observations, with the majority constraining the background mixing ratios. Only around 10 % of the available observations are used for constraining the

5   emissions, whereby the share remains relatively constant through the years. Moreover, it is noticeable that the trace of **KH** is closer to the number of observations during the years in which the smallest numbers of measurements are provided (e.g. 2008 and 2019).

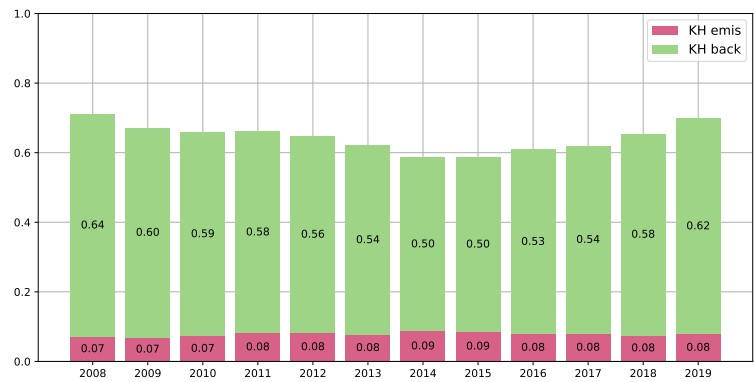

**Figure 8.** Traces of influence matrices divided by the number of available measurements of the corresponding year. See Sect. 2.1 for details on the computation of the influence matrix. The closer $tr(\mathbf{KH})$ is to 1, the more observations are used in the inversion.

With this limited availability of data, a higher percentage of the observations is used as information for the inversion. By contrast, in years during which more observations are available (e.g. 2015), a higher share is identified as noise and hence

10   redundant information, similarly to spatial redundancy in regions where the observation network is denser.

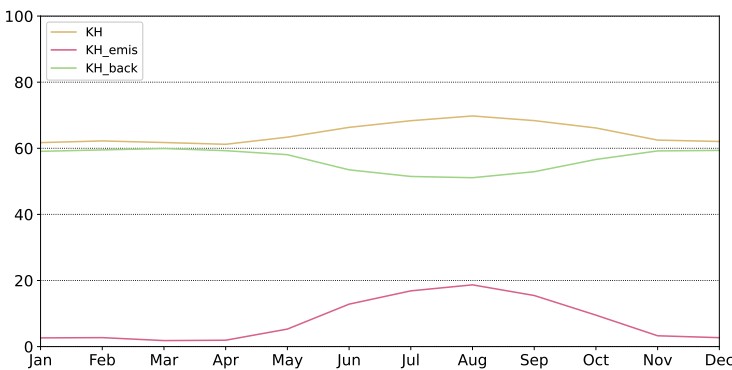

**Figure 9.** Seasonal variation of $tr(\mathbf{KH})$ averaged over the period of interest (2008-2019). The monthly traces are divided by the number of available observations for the corresponding month.





The fraction of useful information in the available observations follows a seasonal variability as shown in Figure 9 (see Figure S3 in the supplements for seasonal variations of the individual years).

The constraints on the emissions during winter are relatively small since the $CH_4$ emissions are comparatively smaller than during the summer months, but also because meteorological conditions (in particular a stratified cold boundary layer) make the

comparison of observations with simulations more challenging. During the summer months, a higher fraction of observations (up to 20 %) is used to constrain emissions. In general, the total trace $tr(\mathbf{KH})$ is higher during the summer month which means that less of the observations are identified as noise. However, additional constraints on the emissions during summer do not ensure constant constraints on the background. Instead, a share from the constraints on the background mixing ratio is transferred to constrain the emissions during the summer months.

By construction $tr(\mathbf{KH_{back}})$ is proportional to $\mathbf{B_{back}}$ and $\mathbf{H_{back}}$. $\mathbf{H_{back}}$ cannot be reduced due to the physics of the atmospheric transport (see Table S4). One way to reduce the share of the information constraining the background in the inversion set-up would be to decrease the uncertainties on the background mixing ratios in $\mathbf{B_{back}}$. This relies, however, also on the performances of simulations of global $CH_4$ concentration fields. Even though in recent years those applications have already improved, they still do not provide a sufficient level of precision that would allow to reduce the uncertainties for the

implementation of our inversion set-up (Inness et al., 2019).

Moreover, the limited transport backwards in time in FLEXPART (10 days in our case) is much smaller than the average residence time of air masses in the Arctic (typically a few weeks; see, e.g., Berchet et al., 2020). Hence part of the influence of Arctic fluxes on observations is diluted in the background in our system. One way of mitigating this issue would be to dramatically increase the backward transport time of virtual particles up to a few weeks; but to limit numerical artefacts, multi-

weeks backward simulations need a very large number of particles to be accurate, at the expense of much high computational costs. Another way of solving the issue would be to fully couple FLEXPART within a global circulation model, thus accounting for the influence of fluxes on observations indefinitely backwards in time; this is what is done in, e.g., Maksyutov et al. (2020) or could be done in the Community Inversion Framework with one of the available global models (LMDZ or TM5; Berchet et al., 2021b).

### 4.2.3   Spatial distribution of constraints on regions and sources

The influence matrix $\mathbf{KH}$ defines how well each emission sector is constrained by the inversion in each sub-region. The majority of the $CH_4$ sources are quite poorly constrained in the sub-regions defined in Section 2.3 with the elements of the influence matrix being less than 10 %. In comparison to that, the wetland emissions are relatively well constrained as shown in Figure 10. Hereby, the figure on the left shows the average constraints over all years, the middle and right figure show two

exemplary years (2011 and 2014) to highlight inter-annual differences. The remaining years are shown in Figure S4 in the supplements.

The average values of the annual influence matrices (Figure 10a) indicate that the current observation network is able to constrains wetland emissions well for most North American sub-regions. In Eurasia on the other hand, most areas are unseen by the inversion and the well constrained areas are predominantly limited to certain parts of Siberia (e.g. the West Siberian





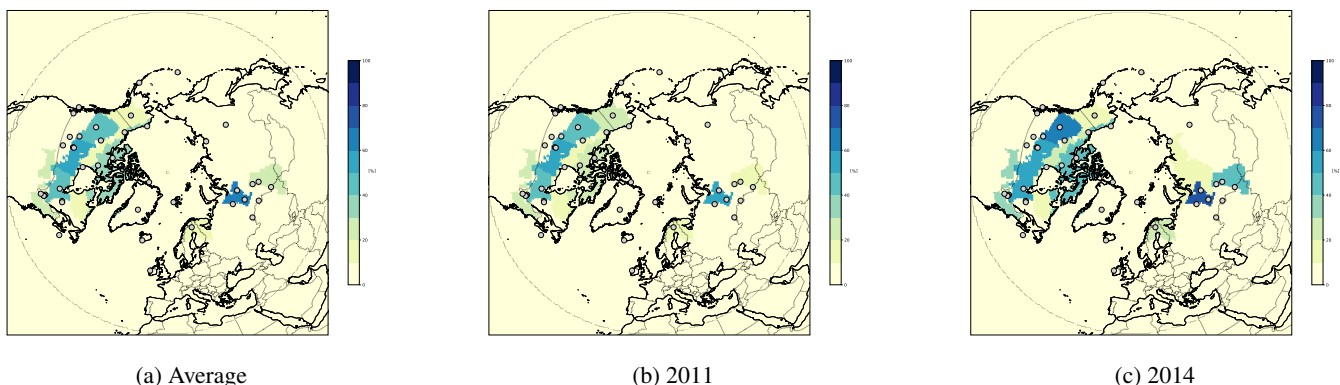

(a) Average           (b) 2011           (c) 2014

**Figure 10.** Regional constraints on wetland emissions. Darker areas indicate a higher percentage of constraints. The observation sites are marked as grey circles.

Plains). This is partly due to the distribution of the observation network (the denser the network, the better the constraints) and to the heterogeneity of data collection within the period of interest (some years have much more available observations than others, especially towards the end of the period). As shown in Figure S4d and Figure S4g, the extend of the constraints strongly varies between the different years due to the availability of observations in Eurasia. Those variations are can also be noticed in

North America, however the well constrained areas remain relatively identical over the whole period.

Another cause of the limited constraints on the emissions is that the available observations in Russia are rather used to constrain the background mixing ratios (see Section 4.2.2). In North America, where a larger number of observation sites are established and more evenly distributed over the area, the observations of certain stations are used to provide the information on the background.

Installing additional observation sites in high northern latitudes in Eurasia would therefore be useful to better constrain local emissions in the future. However, measurement stations in lower latitudes at the sub-arctic boundary would also be necessary to better constrain transport from $CH_4$ hotspots such China, India and the Middle East.

### 4.3 Analysis of posterior fluxes

#### 4.3.1 Total methane fluxes

In order to compare the prior to the posterior fluxes, the area of interest is divided into to four different supra-regions: North America, East Eurasia, West Eurasia and the Arctic (including the High and Low Arctic) as shown in figure Figure 11.

Since most emission sources don't show large differences between the prior and the posterior state and are also poorly constrained by the inversion (Section 4.2.3), the sectors described in Section 3.2.1 are combined to wetlands, other natural (including the $CH_4$ sink from soil oxidation) and anthropogenic emissions. In particular, geological fluxes from the ocean do

not deviate significantly from the prior and are not further commented here. Thereby, the combined natural and anthropogenic





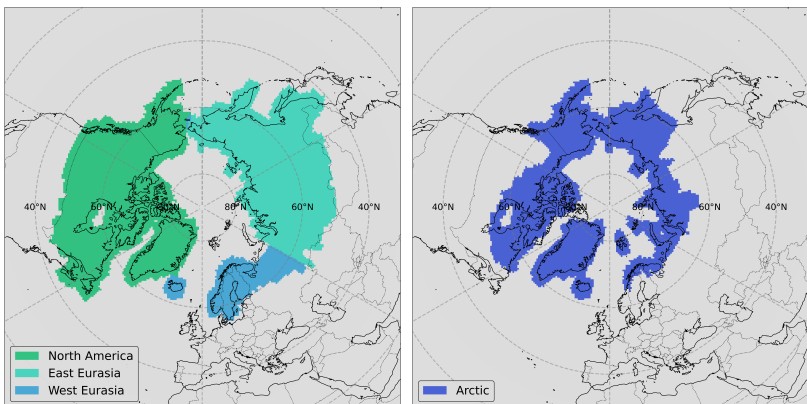

**Figure 11.** Supra-regions for analysis of posterior CH$_4$ fluxes.

fluxes from biomass burning are included in the natural emission sources for simplification since the natural emissions are exceed the anthropogenic ones.

The mean annual prior and posterior CH$_4$ emissions in each region are shown in Figure 12 and, more detailed, in Table S3 for the set-up $\mathbf{x}^a_{max}$ with the highest log-likelihood (Section 2.2.1) together with the corresponding uncertainties obtained

from the $\mathbf{P}^a$ matrix. As expected, the poorly constrained anthropogenic and other natural emissions don't show significant changes between the prior and posterior fluxes for either of the regions, neither in their magnitude nor in their uncertainties. The wetland emissions are decreased in the posterior state, except in West Eurasia. The largest decrease is found in North America, which is also the region best constrained by the inversion. Here, the prior wetland emissions have a magnitude of around 30±26 Tg CH$_4$/year whereas the posterior emissions amount to 19±13 Tg CH$_4$/year. Even though the uncertainties

of the posterior wetland fluxes are still high with around 69 %, they are reduced by around 17 % in comparison to the prior uncertainties. In East Eurasia, the wetland emissions are decreased from approximately 14±12 to 12±10 Tg CH$_4$/year and in the Arctic from 13±11 to 10±8 Tg CH$_4$/year with an uncertainty reduction of respectively 8 and 6 %.

**Comparison to global inversion set-ups**

In order to compare this study to other inversion set-ups, the prior and posterior emissions are set against five different posterior

states obtained with variational inversion frameworks used for the Global Carbon Project (GCP). The comparative CH$_4$ fluxes are hereby an updated version of the results from Saunois et al. (2017, 2020). The variational inversions are performed globally with two different inversion systems, CIF-LMDz using surface observations (Thanwerdas et al., 2021) and PYVAR-LMDz using satellite observations from GOSAT (Zheng et al., 2018). The inversion set-ups 1 and 2 use the prior fluxes distributed for the Global Methane Budget and TRANSCOM chemical fields with the latter including OH inter-annual variability from Patra

et al. (2021). The third set-up is a sensitivity test where freshwater fluxes are added in the prior state. The mean annual total





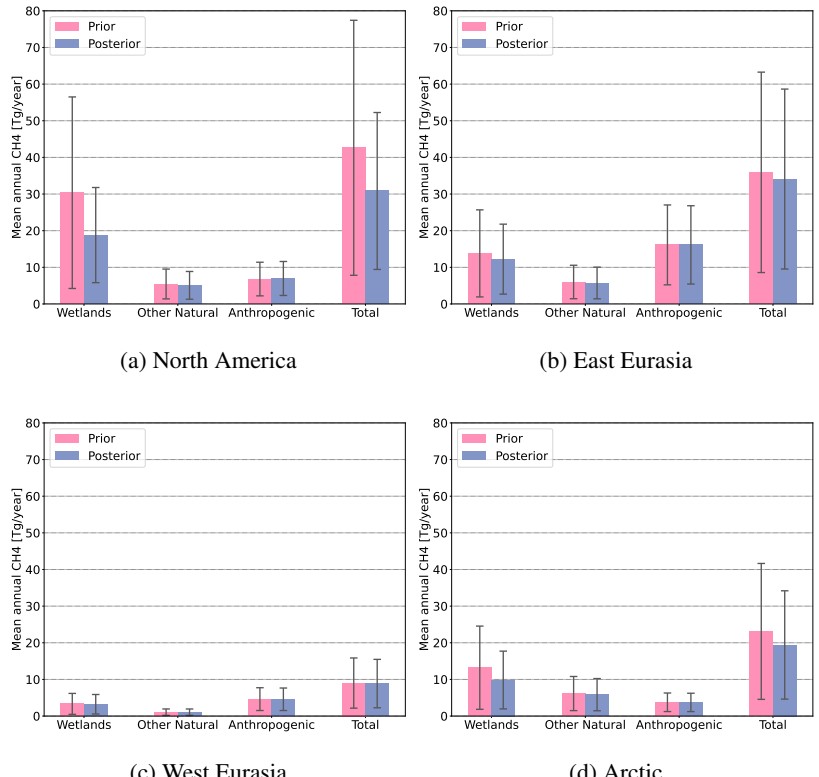

(a) North America            (b) East Eurasia

(c) West Eurasia            (d) Arctic

**Figure 12.** Mean annual $CH_4$ emissions by sector for $\mathbf{x^a_{max}}$ with corresponding uncertainties.

$CH_4$ emissions in the different regions are shown in Figure 13. Since the GOSAT observations are not available for the years 2008 and 2009, the PYVAR-LMDz posterior results are averaged over the remaining period of interest.

In general, the total fluxes of the variational inversion set-ups are all lower than the posterior results of this work. The largest discrepancies are found in the Arctic, where the total posterior fluxes are up to 59 % higher than the results from GCP and

only the inversion set-up number 3 lies within the posterior uncertainty range of our inversion set-up. In North America, the $CH_4$ fluxes of the variational inversion set-ups are between 14 and 44 % lower, in East Eurasia between 38 and 51 % and in West Eurasia between 18 and 38 % in comparison to our posterior emissions. In all of the regions, the results from the inversions using satellite data (PYVAR-LMDz) are the least consistent with the posterior $CH_4$ emissions obtained in this work. The smallest difference to our results is given by the inversion set-up in which the freshwater emissions are added in the prior

state (set-up 3).

As our system explicitly provides posterior uncertainties, contrary to many other inversion systems, it is possible to assess the consistency of our results with other inversions. The discrepancies between the posterior methane emissions from our study and the global variational inversions could be due to the fact that global inverse systems do not perform as well in high latitudes. This has already been identified in Saunois et al. (2017) and can be tracked back to (i) global inversions use fewer observation

sites in the Arctic, (ii) global inversions constrained by satellite measurements have only very few data point above 30°N, (iii)



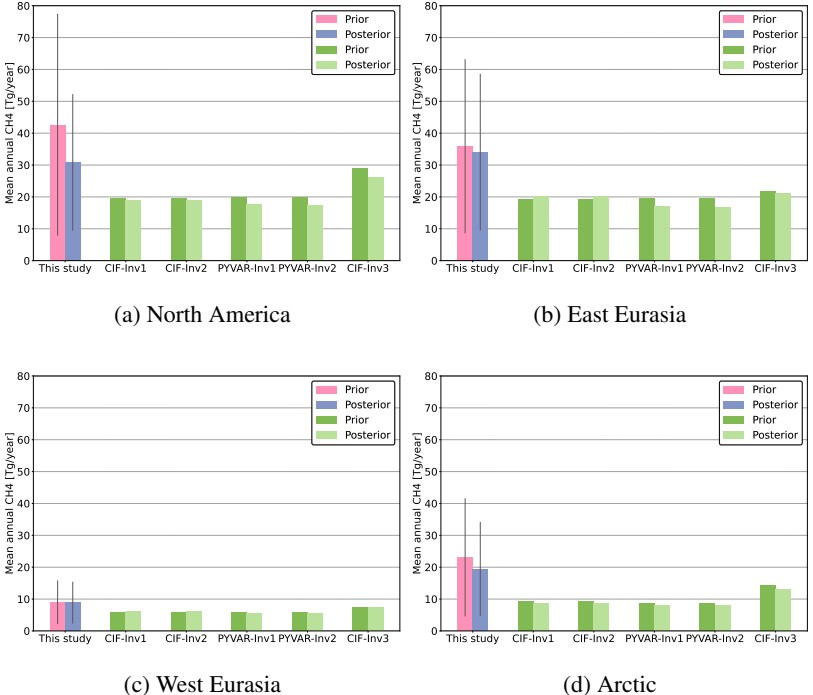

**Figure 13.** Total mean annual CH$_4$ emissions in comparison to different inversion set-ups from GCP.

global models with very low resolution cannot reproduce the Arctic atmosphere properly. However, the discrepancies should be further inquired into.

**Comparison to previous Arctic studies**

In comparison to previous studies using inverse modelling to assess methane emissions in high northern latitude regions our

5 results lie roughly in the same magnitude. Thompson et al. (2017) concluded the total CH$_4$ emissions between 2005 and 2013 to lie between 16.6 and 17.1 Tg/year in North America (above 50°N) and Baray et al. (2021) estimated the combined natural and anthropogenic emissions in Canada at 16.6 and 18.2 Tg/year (between 2010 and 2015). Both values are within the lower limit of the uncertainty range of our ensemble of posterior states in North America (31±15 Tg/year). Berchet et al. (2015) estimated the methane fluxes in the Siberian lowlands to be between 5 and 28 Tg/year in the year 2010 (comparable to region East Eurasia

10 in this study at 34±18 Tg/year). In Eurasia, the total CH$_4$ emissions obtained by Thompson et al. (2017) are between 55.2 and 59.5 Tg/year which is at the higher limit of the uncertainty range of the results from out study for the combined areas of East and West Eurasia (43±23 Tg/year).

Due to the differences in the spatial extent of the regions covered in those studies it is however difficult to obtain reliable comparisons of the estimated methane emissions.



### 4.3.2  Trends of emission sources

In a changing climate, detecting changes in trends of regional emissions in high northern latitudes is critical. Therefore, the trends of all 5000 possible posterior fluxes from the ensemble (see Section 2.1) have been calculated by sector and region. The results for wetland emissions, which is the only source well constrained by the inversion, are shown in Figure 14 for North America and East Eurasia:

- the mean annual CH$_4$ emissions are displayed on the horizontal axis and the corresponding trend of the annual wetland fluxes on the vertical axis.

- the associated probability density functions (PDFs) are shown next to the corresponding axes

- the darker shaded segments show the range of the ensemble $\{x_{max}^a\}$ with the most plausible error configurations (Section 2.2.1) which make out 55 % of the total ensemble.

- the posterior result with the maximum log-likelihood $x_{max}^a$ is highlighted as well as the trend and the mean annual emissions of the prior flux estimates.

Since the data set of the wetland emissions is equal for each year within the period of interest, there is no trend in the prior state. The trend of the posterior wetland emissions in North America (Figure 14a), including all possible uncertainty configurations, ranges approximately between -7.3 and 12.2 %/year with corresponding mean annual emission between around 15 and 30 Tg CH$_4$/year. The trends of the corresponding ensemble of $\{x_{max}^a\}$ range between -1.4 and 1.2 %/year, with 65 % of the 2740 posterior results showing a negative trend. The most plausible of all set-ups $x_{max}$, according to the log-likelihood, also has a decreasing trend of -1.4 %/year. Thus, according to our system, although small (less than 20% per decade), there is a plausible negative (although uncertain) trend on wetland emissions in North America between the years 2008 and 2019.

The trend of the posterior results of the wetland emissions in East Eurasia shown in Figure 14b ranges between -7.5 to 11.7 %/year and mean annual amount of CH$_4$ emissions between 10 to 15 Tg CH$_4$/year. Here, the elements of $\{x_{max}^a\}$ do not include any negative trends with values ranging between 0 and 2.1 %/year and $x_{max}^a$ shows increasing trend of 0.8 %/year. The results point to a very small but statistically significant positive trend in East Eurasia. A positive growth rate in CH$_4$ mixing ratios between 2009 and 2019 in West Siberia was detected by Someya et al. (2020) and attributed to increased wetland emissions in this area, compatible with our conclusion.





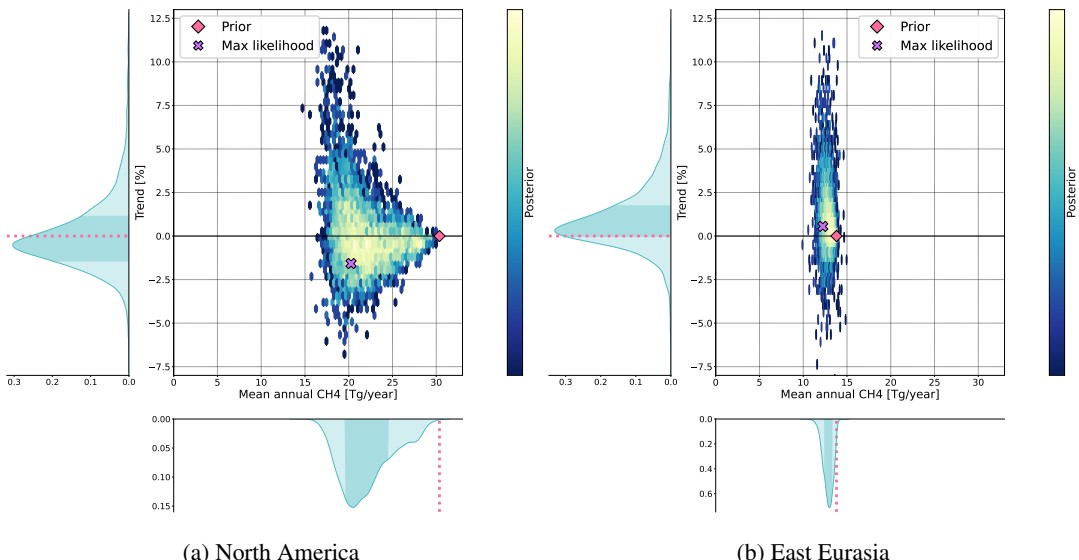

(a) North America                                    (b) East Eurasia

**Figure 14.** Trend and mean annual fluxes of wetland emissions for ensemble of posterior results with corresponding density distributions. Brighter colors of scatters indicate a higher density.

To give an example of a CH$_4$ emission source with a trend in the prior state, Figure 15 shows the emissions from biomass burning of the two beforehand discussed regions. Since the uncertainties on the emissions from biomass burning have been chosen to be higher in comparison to the wetland emissions (Section 3.2), the posterior results contain several negative CH$_4$ fluxes, which are not included in the figures. In both regions, the prior state shows an increasing trend of 30 %/year in North America and 9 %/year in East Eurasia with corresponding mean annual emissions of 1.15 and 2.01 Tg CH$_4$/year.



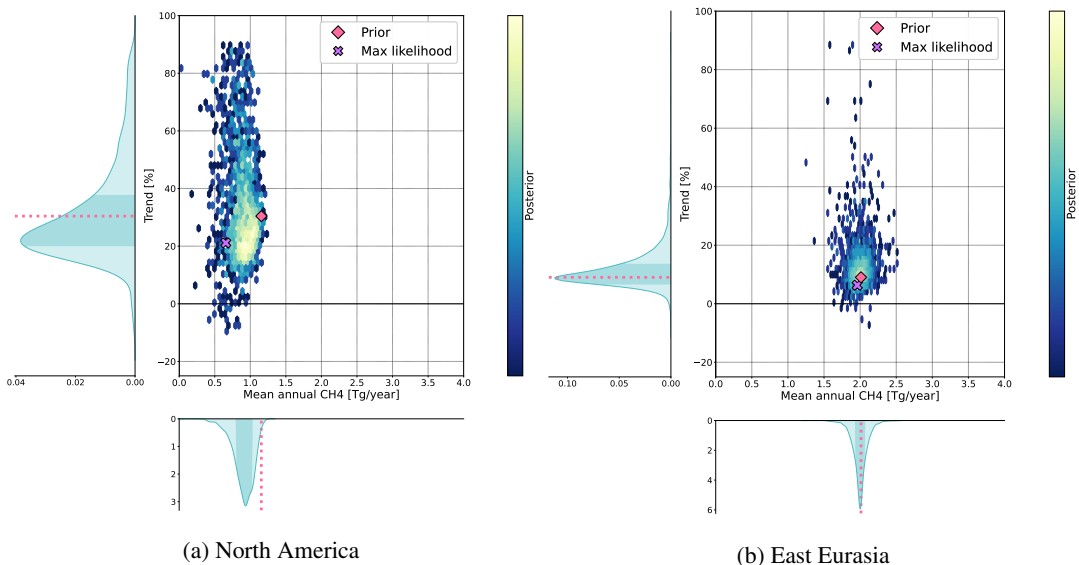

(a) North America
(b) East Eurasia

**Figure 15.** Trend and mean annual fluxes of biomass burning emissions for ensemble of posterior results with corresponding density distributions.

In North America (Figure 15a), the posterior trends show a large variability with values of the total ensemble lying between -11/year and 92 %/year. The trend of the most plausible configuration $x^a_{max}$ is around 21%/year which still indicated a positive trend, however lower than the prior estimate. The magnitude of the $CH_4$ emission from biomass burning in North America is predominantly decreased in the posterior state with $x^a_{max}$ being 0.7 Tg/year lower than the prior mean annual emissions.

In East Eurasia (figure 15b), the mean annual emissions of the total ensemble lie between 0.35 and 3.47 Tg $CH_4$/year and the trends show, equally to North America, high variability in the posterior state (between -8 and 89 %/year). The ensemble $\{x^a_{max}\}$ on the other hand only shows minor deviations and both the trend and the fluxes are close to the prior state.

It has to be mentioned that, even though the fluxes from biomass burning are partially well constrained in some regions and years, the emissions are poorly constrained throughout the whole period of interest and as described in section 4.2.3, the results are highly uncertain.

For most of the methane emissions sources and sinks, the results of the posterior ensemble do not show major deviations from the prior state at all, independent of the prior and observations error.

In addition to that, the obtained trends of the posterior fluxes could as well be influenced by the varying data availability and the the results are therefore still highly uncertain.

### 4.3.3 Seasonal variability

Subsequently, the seasonal cycles of the prior and posterior $CH_4$ fluxes are examined. Figure 16 shows the seasonal cycle of the prior and posterior fluxes for the total $CH_4$ emissions in North America. Hereby, the displayed posterior fluxes are the median





values of the ensemble $\{x^a_{max}\}$. To achieve a better comparison between the different years, the monthly values are divided by the maximum methane fluxes of the prior state.

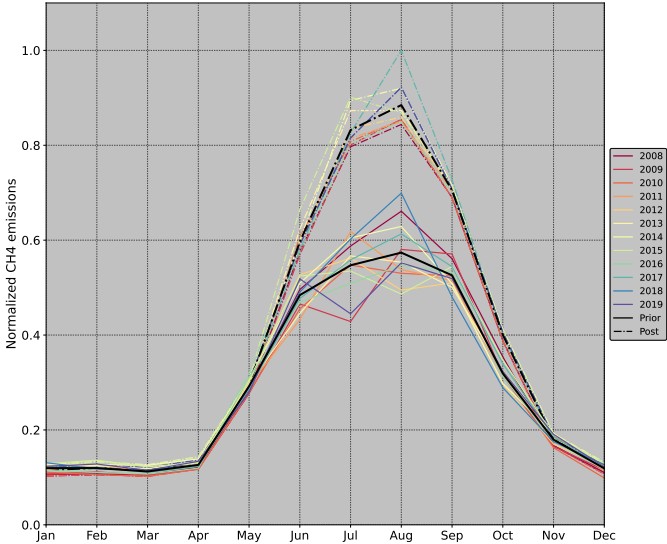

**Figure 16.** Normalized total $CH_4$ fluxes of prior (dash-dotted lines) and posterior (continuous lines) state per month in North America. The coloured lines display the different years, the black lines show the average over all years.

Over the course of the period of interest, the prior emissions show a greater consistency in the annual seasonal cycles. The peak of the emissions are predominantly in August, sporadically already in June. The sectors that contribute to the seasonal

5   differences of the prior state are emissions from wetlands and biomass burning as well as the soil oxidation (section 3.2.1). Since the data set of the wetland emissions and the soil oxidation are consistent for each year, the differences in the seasonality of the prior fluxes is entirely driven by the $CH_4$ fluxes from biomass burning. In comparison to that, the average of the posterior state still reaches the maximum emissions in August, however the peak is less pronounced and the emissions decrease more gradually during the autumn months. The annual seasonal cycles of the posterior fluxes are more divergent from each other.

10   The majority of the years still show the highest methane emissions in August, although some years (e.g. 2012 and 2015) have a local minimum during that month. Unlike the prior state, the differences in the seasonal fluxes of the different years are not exclusively influenced by emissions from biomass burning. As shown in figure 17, most of the changes in the seasonal cycle of the total $CH_4$ emissions arise from adjustments of the monthly wetland fluxes since the local high and low points are predominantly during the same time of the year.



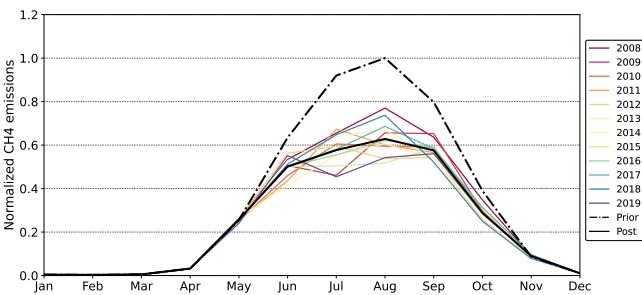

**Figure 17.** Normalized wetland CH$_4$ fluxes of posterior state per month in North America. The black continuous line shows the average prior, the dash-dotted line the average posterior state.

The other three regions assessed in this study show similar minor changes in the seasonal cycles which are predominantly influences by wetland emissions and don't show any clear seasonal pattern between the different years.

### 4.3.4 Inter-annual variability

Finally, we inquire into the inter-annual differences of the methane emissions. Figure 18 shows a time series of the total
CH$_4$ fluxes in prior and posterior state. The posterior emissions are again obtained by calculating the median of the most plausible results $\{\mathbf{x}^{\mathbf{a}}_{\mathbf{max}}\}$, with the corresponding minimum and maximum values of this ensemble marking the uncertainty range. Therefore, in the following section, the given quantification of the monthly CH$_4$ emissions refers to the median values.

In West Eurasia (figure 18c) there is effectively no inter-annual variability, neither in the prior nor the posterior state, with peak values of the 1.2 Tg CH$_4$/month each year; small deviation from the prior comes from very small constraints on that
region. Similarly, the emissions in the Arctic (figure 18d) show very little deviations in the prior state, with the maximum fluxes only deviate by 0.1 Tg. The posterior fluxes on the other hand gradually decrease from a peak value of 4.0 CH$_4$/month in 2008 to 3.1 CH$_4$/month in 2015, which marks the year with the largest difference to the prior state. In the following years the emissions slightly increase again to a maximum value of 3.3 Tg/month and stay constant until the end of the period of interest. In East Eurasia, shown in figure 18b, the inter-annual variations are fairly low both in the prior and posterior state. According
to the prior estimates, the highest methane emissions occur in 2012 with a peak value of 5.6 CH$_4$/month. Also, the fluxes are slightly lower in the beginning of the period of interest (around 4.9 Tg) than during the last four years (around 5.4 Tg). The posterior emissions mostly follow that pattern with a maximum reduction of 0.5 Tg in 2012. The most prominent variabilities are, unsurprisingly, found in the North America (figure 18a) which is best constrained by the inversion. As mentioned before, the inter-annual differences of the prior state are caused by variations in the emissions from biomass burning. The largest prior
total methane emissions occur in 2017 with a maximum of 9.5 Tg CH$_4$/month, the year with the lowest peak emissions (7.9 Tg CH$_4$/month) is 2008. Like the emissions in East Eurasia, the peak values of the prior state are generally higher during the last years of the period of interest whereas the years with the lowest CH$_4$ emissions can be found at the beginning. The posterior fluxes in North America vary between peak emissions of 5.0 Tg (2015) and 6.6 Tg (2018) without showing a clear pattern.



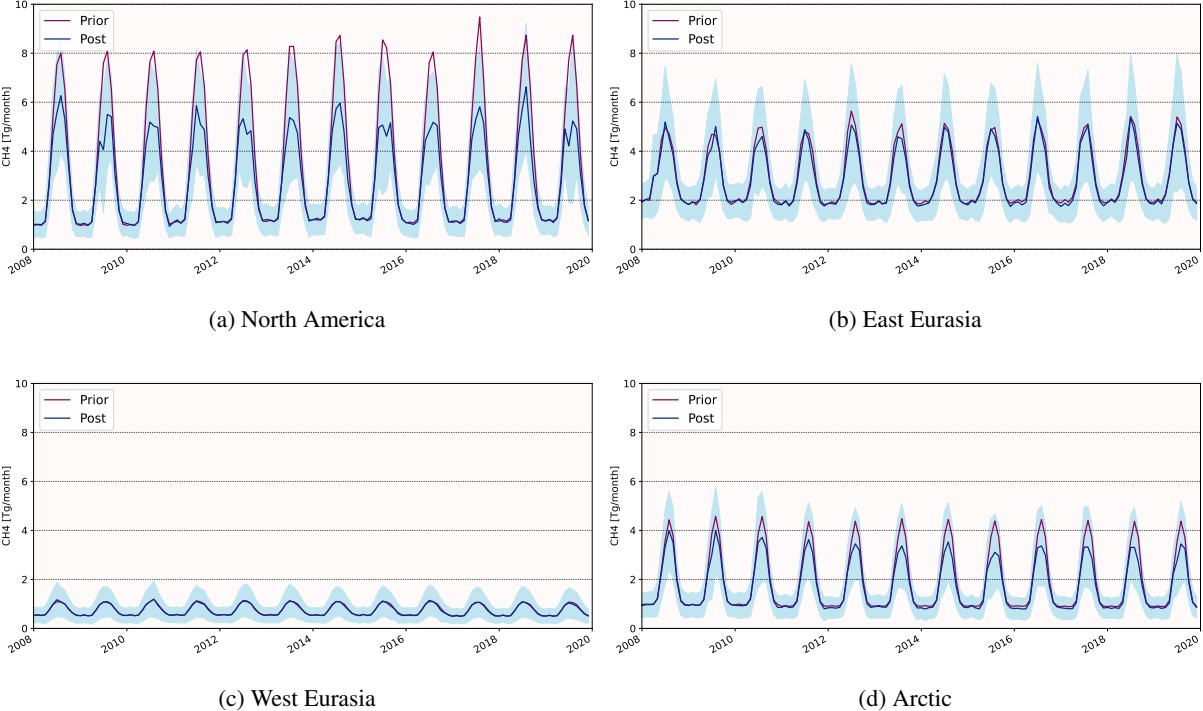

**Figure 18.** Total seasonal prior (purple) and posterior (blue) $CH_4$ emissions between 2008 and 2019.

In fact, the year 2008, which shows the lowest emissions in the prior state, has the second highest methane fluxes of all years whereas the emissions in the year 2019 are almost as low as in 2015.

An explanation for the large discrepancies in the inter-annual variabilities between the prior and the posterior state in North America is the reduction of fluxes from biomass burning in the posterior state which are shown in figure 19. The prior estimates show a large variability with exceptionally high emissions in the years 2013 to 2015 and, most evident, in the year 2017 with up to 1.4 Tg $CH_4$/month. These increased emissions during certain years do not agree with the observations though which is why the peak emissions in the posterior state are up to 50 % lower.



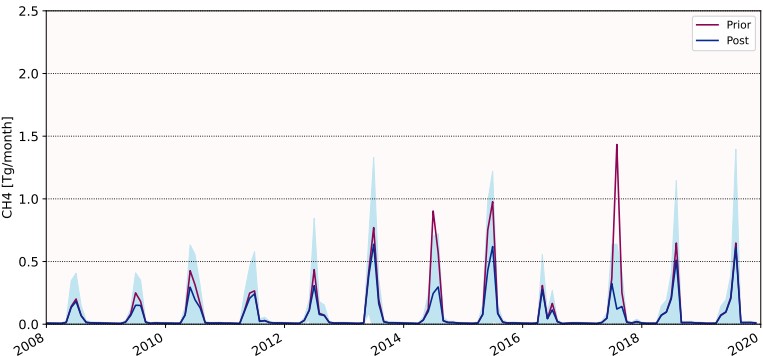

**Figure 19.** Seasonal $CH_4$ emissions from biomass burning between 2008 and 2019.

Another factor that has to be taken into account when analysing the posterior inter-annual variabilities is the inconsistent availability of observations throughout the different years. For years during which only few information from the measurements are available, the posterior state is likely to be closer to the prior.

## 5  Conclusions

We designed an inversion system to constrain $CH_4$ surface fluxes in high northern latitudes based on surface observations in Arctic regions and an atmospheric transport model. Extensive sensitivity tests were carried out to comprehensively assess the methane emissions and uptake, respectively, from different $CH_4$ sources and sinks during the years 2008 to 2019. We aimed to reduce uncertainties on current bottom-up estimates and thereby gain a more accurate understanding of the the extent, seasonality and inter-annual trends of methane emissions in Arctic and Sub-Arctic regions. In order to achieve that, we computed a total of 5000 posterior states of posterior methane fluxes with varying uncertainties on the observations, background mixing ratios and prior flux estimates and evaluated their plausibility to get a reliable assessment of the methane emissions in high northern latitudes.

The atmospheric observations used for this study included both quasi-continuous and discrete measurements from 41 observation sites in different Arctic nations. We found that this observations network is not sufficient to satisfactorily constrain most $CH_4$ sources and sinks in high northern latitudes over the whole period of interest. Only wetland emissions are adequately constrained in North America and sporadically in Russia although with inter-annual variabilities. It is therefore not possible to reduce uncertainties on most $CH_4$ emissions sources and sinks occurring in high northern latitude regions to a substantial extent. Besides, a considerable share of the observations is used by the inversion to constrain the background mixing ratios. This share could be reduced by improved initial $CH_4$ mixing ratio fields which would allow for lower uncertainties on the background. Moreover, additional stations at the sub-arctic boundary would be necessary to better constrain transport from $CH_4$ hotspots such China, India and the Middle East. If adding stations in the buffer zone, which in this case included latitudes from 30 °N, would improve the constraints on high northern latitude regions should be investigated further.

The obtained posterior $CH_4$ fluxes were, in comparison, predominantly lower that the prior estimates though still higher than comparable posterior results from variational inversion set-ups. In North America, the average total methane fluxes were reduced by around 11 Tg/year with a corresponding uncertainty reduction of 26 %. In East Eurasia and the Arctic, fluxes were declined by 2 and 3 Tg/year, respectively, with uncertainty reductions of 13 (East Eurasia) and 12 % (Arctic).

Significant changes in the seasonal cycles of the methane emissions could not be observed in either of the regions studied. Minor shift in the seasonal cycles in certain years were exclusively influenced by $CH_4$ emissions from high-latitude wetlands.

Inter-annual differences were most significant in North America where the largest discrepancies between the prior and posterior state could be observed since the region is well constrained. Whereas the highest peak emissions in the prior state took place at the end of the period of interest, raised methane emissions in the posterior state were predominantly observed

at the start. The $CH_4$ peak emissions were hereby also up to 4.3 Tg/month lower. Those differences can be explained by a reduction in emissions from biomass burning, which appear to have been overestimated for certain years (e.g. 2014 and 2017) in the prior estimate.

The wetland emissions in North America showed a small decreasing trend between 2008 and 2019 whereas the $CH_4$ emissions from wetlands in East Eurasia were slightly increasing within the period of interest. Since most regions in the Arctic and

Sub-Arctic were poorly constrained by the inversion, most methane emissions sources as well as the soil oxidation didn't show any significant trends in the period under study.

To get a conclusive understanding about magnitude, long-term trend and seasonal variability of methane emissions in the entire Arctic region, it would be beneficial to expand the observation network, especially in Eurasia, to better constrain the area for future works. Complementary approaches bringing fixed and mobile platforms (ships, aircrafts, trains, etc.) together should

also be explored to refine our understanding of the regional Arctic budget (e.g., Pankratova et al., 2022; Berchet et al., 2020; Pisso et al., 2016; Thornton et al., 2020, etc.). New satellite platform may also in the future expand our coverage of Arctic methane emissions, even though technical difficulties (albedo, clouds, etc.) hampers our capability to use high-latitude satellite retrievals.

This is particularly important since ongoing environmental changes due to rising temperatures in high northern latitudes are

affecting natural sources and sinks of $CH_4$, further complicating the estimation and prediction of Arctic methane emissions, their contribution to the global budget and the resulting potential climate feedback.

*Code and data availability.* The transport model FLEXPART is open-source and can be downloaded here: flexpart.eu. The meteorological forcing fields for FLEXPART are interpolated from open ERA5 re-analysis, extracted using the open-source flex-extract toolbox (Tipka et al., 2020, flexpart.eu/flex_extract; last access: 14/11/2022). Flux data were obtained from the Global Carbon Project - $CH_4$ (icos-cp.eu/

GCP-CH4/2019; last access: 14/11/2022). The contribution of the background concentrations was calculated using the Community Inversion Framework available here: Berchet et al. (2021a).

Observations data from JR-STATIONS network can be obtained upon request to Motoki Sasakawa. Observations from the ECCC network can be obtained upon request to Doug Worthy. Observations from the NOAA-GML network can be downloaded from the dedicated ObsPack web server: gml.noaa.gov/ccgg/obspack/ (last access: 14/11/2022).



*Author contributions.* SW designed the analytical inversion system, run the FLEXPART simulations to compute footprints, performed the scientific analysis presented in the paper. AB and IP provided scientific, technical expertise and contributed to the scientific analysis of this work. MS and AM provided the $CH_4$ fluxes and inversion results from GCP. JT computed the LMDZ simulations used as background for the study. TM, MS, MA, DEW and XL provided measurement data from the NIES JR-STATION network as well as scientific advice. RLT and ES contributed to the code of the FLEXPART interface to the Community Inversion Framework used to compute contributions from the background, as well as scientific advice.

*Competing interests.* The authors declare that they have no conflict of interest.

*Acknowledgements.* We thank all PIs and supporting staff for deploying, maintaining and making available data from numerous (and often remote) observation sites around the Arctic. In particular, we thank our colleagues from the V.E. Zuev Institute of Atmospheric Optics (Tomsk). We thank the data infrastructure of ICOS (Integrated Carbon Observing System; .icos-cp.eu), the WDCGG (World Data Center for Greenhouse Gases; https://gaw.kishou.go.jp/) and ObsPack (gml.noaa.gov/ccgg/obspack/) for distributing observations. We thank F. Marabelle (LSCE) and the LSCE IT team for the computer resources. Sophie Wittig was supported by the CEA NUMERICS program, which has received funding from the European Union's Horizon 2020 research and innovation program under the Marie Sklodowska-Curie grant agreement No 800945.



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
