# Peer review of "Estimating Methane Emissions in the Arctic nations using surface observations from 2008 to 2019"

_EGUsphere, 2022_

## Author Response (AR1)

We thank the reviewers for their time reviewing our manuscript. We are thankful for their insights and fruitful comments helping to improve our work.
We reply below to individual comments.
Reviewers comments are reported in bold, our replies in regular font, while modifications in the manuscript are written in italics.

**Report 1**

**p4, line 17: There is no OH sink taken into account explicitly. However, the impact of atmospheric sinks is implicitly present via the background concentrations. What is the important configuration setting that allows to neglect the explicit impact of sinks, is that the length of the time window of 10 days for the trajectories?**

The justification is mentioned in section 3.3, page 15, line 9 to 13: "Chemical reactions with free radicals in the atmosphere are neglected since the air masses in the studied domain change rather quickly (up to 2 months) compared to the lifetime of CH4 molecules (≈9 years)."
We simulate backward transport over a 10-day period, which means that a peak of 100 ppb caused by emissions, would be oxidized by a rate of less than 0.5 ppb after 10 days, hence negligible for our computations.
It should be noted that the OH sink is accounted for in the global simulations used to compute the background.
We clarify the latter aspect in the manuscript.

In Sect. 2.1 we added:

*We neglect chemical oxidation of CH4 emitted in our regional domain by OH as further explained in Section 3.3, although oxidation by OH is still accounted for in the global model used for the background (Section 3.3.3).*

The header of Sect. 3.3 was modified to:

*The oxidation of methane by hydroxyl radicals (OH) is neglected inside the domain of simulation since the lifetime of CH4 is ≈9 years (Prather et al., 2012) and the air masses remain in the domain up to 2 months (Berchet et al., 2020). The methane sink from the hydroxyl radical is however accounted for in the global simulations used to compute the background mixing ratios (Section 3.3.3).*

**2.2.1 Pairs of $(R,B)$ are judged on their likelihood. How likely is the configuration with the highest log-likelihood actually? This might be examined by comparing a histogram of the ratios between elements of $S^j$ and std. deviations from the diagonal of $S^j_{RB}$ with the standard normal distribution. Part of this is discussed in Section 4, but it would be useful to have some insight here already.**

We agree that the log-likelihood method tries to attribute diagonal terms of $S^j_{RB}$ as close as possible to the diagonal terms of $S^j$, while keeping the ln | $S^j_{Rji,Bji}$| as small as possible, depending on the degrees of freedom available. Still, we find it

difficult to comment directly on the diagonal terms of these matrices as their ratio is very sensitive to the definition of the problem, to the observations, to the parameters optimized by the log-likelihood, etc.

Nevertheless, the evaluation of the use of the log-likelihood method in atmospheric inverse modeling approaches has been subject of other studies and is not a specific aim of this work. The selection of this method to evaluate the error estimates of matrices of the ensemble was therefore based on other studies concerning this topic.

We added the following references analysing further the log-likelihood method:

*(Winiarek et al., 2011; Berchet et al., 2013, 2015a)*

**3.3.2 and 3.3.3, Eq. (7) and (9): Does this actually provide a mathematically correct covariance matrix? Usually a covariance is formulated as the matrix product $DCD$, with $D$ a diagonal matrix with the std. deviations, and $C$ a correlation matrix with elements following for example the exponential decays in these equations. The leading factor would then be $\sigma_m\sigma_n$ rather than the average shown here.**

The equation was indeed incorrectly written and has been corrected.
As mentioned below, the accounting of background covariances should anyway be improved in a future work to include the 4D background fields in the control vector instead of the background contribution time series by station.

**3.3.2 and 3.3.3: The correlation lengths $d_{corr}$ and $t_{corr}$ are kept fixed. Would it in the current setup of the experiment be possible to also perturb these in the ensemble? If so, what implications would that have for the ensemble size? With longer correlation lengths the number of degrees of freedom will be come lower, and maybe more in line with the number of observations.**

Drawing Monte-Carlo members by varying d_corr and t_corr is perfectly possible. We agree that larger d_corr and t_corr reduce the number of degrees of freedom relative to the background. Inquiring the likelihood of configurations relative to d_corr and t_corr would be an interesting new axis of work; however, we limited our present work to fixed d_corr and t_corr to keep the message clearer and also because in our set-up the background takes the shape of time series by station; too strong correlations between stations may be counter-productive in terms of inversions as nearby stations can be influenced by different background air masses.

A next step will be to transform our observation operator to include global 4D fields as the background instead of their extraction/convolution at given sites accounting for backward transport. This would allow to realistically link background from different stations by accounting by the non-isotropic transport.

We modified the end of Sect. 3.3.3 to comment on that aspect:

*Drawing Monte-Carlo members by varying dcorr and tcorr is perfectly possible and would impact the number of degrees of freedom relative to the background (the*

*higher dcorr and tcorr, the smaller the number of degrees of freedom). In the present work, we limit our analysis to a single set of dcorr and tcorr. The chosen configuration is a compromise to account for the consistent influence of the background between neighbouring stations and successive time steps while avoiding forcing unrealistic isotropic correlations when close sites are influenced by different background air masses. A future work will include the four-dimensional background fields in the control vector as the background.*

**Section 3.3.2 How is the "near-surface residence time" defined? That is, at which heights above the surface are the particles receiving emissions? I guess that a layer structure is defined, with a model for how emissions are distributed over these layers; could this be clarified?**

We defined a height threshold below which particles are considered as influenced by surface fluxes. We do not distribute surface fluxes into different layers. This is a known caveat for biomass burning in particular that should account for an injection height.

We add in Sect. 3.3.2:

*The near-surface hereby corresponds to particles below 500 m instead of, for instance, the planetary boundary height (PBL) since simulated PBL in the Arctic can be unrealistically small, especially during the winter months.*

**p 16, line 23: Isn't "EAC4" is simply the ECMWF experiment ID, rather than an abbreviation?**

"EAC4" is in fact an abbreviation for "ECMWF Atmospheric Composition Reanalysis 4" as also described in the corresponding reference (Innes et al. 2019, section 3.3.3, page 16, line 24).

**Report 2**

**p.4 l 17. Neglecting the OH sink seems a little unexpected, because of its importance, trends and seasonality. What where the challenges in applying it ? How would it affect the results ?**

The justification is mentioned in section 3.3, page 15, line 9 to 13: "Chemical reactions with free radicals in the atmosphere are neglected since the air masses in the studied domain change rather quickly (up to 2 months) compared to the lifetime of $CH_4$ molecules (≈9 years)."
We simulate backward transport over a 10-day period, which means that a peak of 100 ppb caused by emissions, would be oxidized by a rate of less than 0.5ppb after 10 days, hence negligible for our computations.
It should be noted that the OH sink is accounted for in the global simulations used to compute the background.
We clarify the latter aspect in the manuscript

(see 1[st] comment by reviewer #1)

**p.12 l.17 Do the wetland emissions include freshwater emissions, and are they optimized?**

The wetland emissions dataset by Poulter et al. explicitly aims at excluding $CH_4$ emissions from freshwater systems such as lakes and rivers. In high northern latitude regions the distinction is however challenging, since ponds in permafrost peatlands and thermokarst lakes are usually shallow (less than 1 m depth), which falls of the definition commonly used for wetlands (standing water up to 2 – 2.5 m depth). $CH_4$ emissions from freshwater systems are therefore not optimized taken into account as a separate source.

We comment on that aspect in Sect. 3.2.1:

*Terrestrial freshwater systems other than wetlands are hereby not taken into account as a separate source since ponds in permafrost peatlands and thermokarst lakes are usually shallow (less than 1 m depth), which falls under the usual definition for wetlands with standing water up to a depth of 2 to 2.5 m (e.g. Tineret al., 2015).*

**p.14 l.20 What would be the effect of changing these uncertainty assumptions (50%, 60%,100%), as they usually are not very well defined?**

The given uncertainty assumptions only serve as reference values to compute 500 different sets of error estimates, as described in section 3.2.2 on page 14 and 15. The chosen log-normal random distribution of uncertainties leads to a higher number of uncertainty configurations centered around these reference values, yet a large number of different uncertainty estimates are included for each $CH_4$ emission source.

Moreover, we attribute a different weight in our posterior analysis of inversion results depending on the likelihood of the matrices R and B. Hence our posterior estimates account for the most likely uncertainties.

We clarify this aspect in Sect. 3.2.2:

*We expect little sensitivity of our results to these prescribed reference values as we weight every sample according to its log-likelihood (see Sect. 2.2.1).*

**p.15 l.10 Observations are averaged over quite a long time, one month. How would the results change if a shorter time period was used? Would it affect the share of the observed signal used for optimization of background vs regional emissions?**

The share of information used to optimize background versus emissions is presented in Figure 8, page 21. Only less than 10% of available pieces of information is used to optimize emissions, while 25-40% is lost in noise, the rest is used for the background. Using daily observations instead of monthly data could reduce the noise share, and also move part of the background contribution to the emissions. However, there are still obstacles to such ideal outcome:
1) the used background mixing ratio estimates were predominantly too high and therefore a large share of observations was used by the inversion to constrain the background anyway.
2) Observations and simulations at the daily scale are predominantly constituted of alternating peaks due to emission plumes; however, simulated and observed plumes are not necessary synchronized. Using daily observation with temporal mismatches between the numerical realm and the real world would actual worsen usable information. This is particularly critical as we have diagonal observation error covariance matrices R, as is usually done in the inversion community.
3) increasing the duration of backward transport (from 10 days to several weeks) could mechanically reduce the share background information; however, this would critically increase the computation time of footprints and was not affordable for our exploratory study; future work may try inquiring this issue.

We comment on that aspect in Sect. 4.2.2

*Another option to increase the ratio of information used to constrain emissions instead of the background would be to use higher temporal resolution for the observations. A compromise would be to use daily afternoon averages instead of monthly averages. However, as illustrated in Belikov et al. (2019), Arctic observations, even at the daily scale, can see strong daily peaks due to emissions in combination with meteorological conditions. Such peaks may be ill-reproduced by the model and could be shifted in time and magnitude, posing a double-penalty effect to the inversion (Vanderbecken et al.). This can have a critical impact on the inversion conclusion, especially with diagonal observation covariance matrices R, as done in the present work, consistent with to the general practice of the inversion community.*

We also add some comment in the conclusion:

*Increased observation resolution instead of the monthly means used in the present work should be explored as well to potentially improve the constraints on fluxes, although at higher temporal resolution, missed emission peaks in simulations compared to observations could lead to depreciated emission estimates.*

**p.16 l.6 Did you use boundary layer height for defining near-surface conditions?**

Near-surface corresponds to particles below 500 m. We use such a minimal threshold for the surface influence instead of the planetary layer because simulated PBL heights in the Arctic can be unrealistically small, especially in winter, which would cause some periods to virtually see no influence from the surface.
It can be proven that if assuming that the PBL is perfectly mixed, using 500m instead of the PBL height is equivalent when the PBL is higher than 500m. As mentioned above for PBL lower than 500m, this is not mathematically equivalent, but is preferred to avoid numerical artifacts due to erroneously represented winter stable PBL, and also to insufficient number of virtual particles.
We add in Sect. 3.3.2:

*The near-surface hereby corresponds to particles below 500 m instead of, for instance, the planetary boundary height (PBL) since simulated PBL in the Arctic can10be unrealistically small, especially during the winter months.*

**p.16 l.25 The implementation used for the inversion? Maybe some more details here?**

We propose to clarify to:

*The practical computation of background time series from background fields and backward trajectories is carried out using the Community Inversion Framework, …*

**p.18 l.12 Presume the climatological priors are still realistic in their own framework. Maybe rephrase 'not good enough as year-to-year changes were present' or equivalent**

The proposed change has been implemented as following:
*This confirms that the climatological priors are not good enough as year-to-year changes were present and the inversion can realistically improve the flux trends.*

**p.25 l.15 Satellite measurements (e.g. TROPOMI) do have plenty of data points above 30N, especially during summer, extending to high Arctic. Please rephrase.**

We propose to clarify to:

*(ii) global inversions constrained by satellite measurements (GOSAT, IASI) provide fewer data points above 30°N compared to regions in lower latitudes because of factors such as the solar zenith angle, the surface albedo and the limited light during the polar night; more recent inversions start using TROPOMI data (Tsuruta et al.,*

*2023), with possibly a higher density of data points at high latitudes, but these data were not publicly available at the time of the present work,...*

**p-28 l.3 Are the negative fluxes only removed from the figure, or also from the calculation of the trend?**

The trend and mean annual emissions from biomass burning were calculated for each inversion set-up individually. Configurations with negative mean annual emissions of CH4 were subsequently removed from the ensemble, therefore the trends of configurations with negative mean annual CH4 fluxes are not shown in the figures.